# Lifelong Domain Adaptation via Consolidated Internal Distribution

**Mohammad Rostami**
USC Information Sciences Institute
Los Angeles, CA 90007
rostamim@usc.edu

## Abstract

We develop an algorithm to address unsupervised domain adaptation (UDA) in continual learning (CL) settings. The goal is to update a model continually to learn distributional shifts across sequentially arriving tasks with unlabeled data while retaining the knowledge about the past learned tasks. Existing UDA algorithms address the challenge of domain shift, but they require simultaneous access to the datasets of the source and the target domains. On the other hand, existing works on CL can handle tasks with labeled data. Our solution is based on consolidating the learned internal distribution for improved model generalization on new domains and benefiting from *experience replay* to overcome catastrophic forgetting.

## 1 Introduction

Deep neural networks relax the need to for manual feature engineering by learning to generate discriminative features in an end-to-end blind training procedure [1, 2]. Despite significant advances in deep learning, however, robust *generalization* of deep neural network on unseen data is still a primary challenge when *domain shift* exists between the training and the testing data [3, 4]. Domain shift is a natural challenge in *continual learning* (CL) [5, 6] where the goal is to learn adaptively and autonomously when the underlying input distribution drifts over extended time periods. Distributional shifts over time usually lead to performance degradation of trained models which in turn necessitates model retraining to acquire knowledge about new distributions. Current CL algorithms mainly consider tasks, i.e., domains, with fully labeled datasets. Hence, these algorithms require annotating massive training datasets for new observed domains. Persistent manual data annotation, however, is practically prohibitive because of being an economically costly and time-consuming process [7].To relax this constraint, our goal is to develop an algorithm for continual adaptation of a model for tackling the challenge of domain shift in a CL setting using solely unannotated datasets.

Unsupervised Domain adaptation (UDA) is a highly relevant learning setting to our problem of interest. The goal in UDA is to train a model for a *target domain* with unannotated data by transferring knowledge from a related *source domain* in which annotated data is accessible [3]. A primary group of UDA algorithms map the training data points for both domains into a shared latent embedding space and align the distributions of the source and the target domains in that space [8]. Hence, a source-trained classifier that receives its input from the shared embedding space would generalize on the target domain as well. The domain alignment procedure has been implemented either using generative adversarial learning [9, 10, 11, 12, 13] or by directly minimizing the distance between the two distributions [14, 15, 16, 17, 18]. Existing UDA algorithms are not suitable for continual learning because the underlying model can be trained if datasets from both domains are accessible. Moreover, these methods usually consider only a single target domain and a single source domain. Finally, simply updating the underlying model to generalize in the current encountered domain is not sufficient. Because upon updating the model, the network likely would forget the past learned domains as the result of retroactive interference, referred as the phenomenon of *catastrophic forgetting* [19, 20] in the

35th Conference on Neural Information Processing Systems (NeurIPS 2021)

CL literature. Consequently, we also need to tackle catastrophic forgetting in our unexplored learning setting. Our method can be considered as an improvement over existing UDA and CL methods.

**Contributions:** we develop an algorithm for lifelong unsupervised adaptation of a model on new domains using solely unannotated data. Our idea is based on consolidating the internally learned distribution that encodes the learned knowledge by the model when the initial source domain is learned. We use this multimodal distribution to update the model such that the learned distributions for all subsequent unannotated domains are coupled. To overcome catastrophic forgetting, we store important representative samples for all tasks and replay them back when the model is updated. We provide a theoretical analysis to demonstrate that our method mitigates catastrophic forgetting and also leads to improved generalization. We validate our method using standard UDA benchmarks.

## 2 Background and Related Work

Lifelong UDA setting lies on the intersection of both UDA and CL learning settings.

**Unsupervised domain adaptation:** an effective approach for domain alignment in UDA is to use a probability distance to measure distributional discrepancy and then train an encoder to minimize the cross-domain distance at its output as a latent shared embedding space [21, 22, 23, 24, 25, 26]. The Wasserstein distance (WD) [27] is a suitable metric for this purpose due to possessing non-vanishing gradients. This property is beneficial because deep learning optimization problems are primarily solved using gradient-based optimization techniques [23]. In this work, we rely on the sliced Wasserstein distance (SWD) [24] variant of the Wasserstein distance because it can be computed efficiently using a closed form solution via empirical samples of the data distribution.

**Continual learning:** Existing CL methods primarily use either model regularization or experience replay to tackle catastrophic forgetting. The idea of model regularization [20, 5] is to identify the network weights that contribute significantly in retaining knowledge about a past learned task and then consolidate them when the model is updated to learn the subsequent tasks. Alternatively, the model can be expanded progressively to learn the new tasks using added weights [28]. We rely on experience replay which is implemented using pseudo-rehearsal [19]. The idea is to identify important training data points that contribute significantly to learning a task and store them in a memory buffer [29]. These samples then would be replayed along with the current task's data to represent the past learned distributions [30, 31] to retain the acquire knowledge about the past tasks. The major challenge is to identify the important data points. We rely on the consolidated internal distribution to identify the important data points. To mitigate the sample selection challenge, some CL algorithms rely on generative experience replay, where the memory buffer is replaced with the ability of generating synthetic pseudo-samples that are similar to the samples of the past learned tasks [31, 32, 33, 34].

**Domain Adaptation in Continual Learning Settings:** Existing works in this direction are highly limited. Wulfmeier et al. [35] study addressing gradual shifts in changing environments. Bobu et al. [36] and Wu et al. [37] explore addressing the problem of domain shift in continual learning settings but both works assume that datasets for all learned tasks are observable at each time-step. This assumption limits practicality of these works. Porav et al. [38] explore a setting similar to our setting but the proposed method relies on image translation which makes the work highly domain specific. In our work, we relax these limitations and develop a more general algorithm.

## 3 Problem Statement

Consider a classification problem in a source domain $\mathcal{S}$ with a labeled training dataset $\mathcal{D}_{\mathcal{S}} = (\boldsymbol{X}_0, \boldsymbol{Y}_0)$, where $\boldsymbol{X}_0 = [\boldsymbol{x}_1^0, \ldots, \boldsymbol{x}_N^0] \in \mathcal{X} \subset \mathbb{R}^{d \times N}$, $\boldsymbol{Y}_0 = [\boldsymbol{y}_1^0, ..., \boldsymbol{y}_N^0] \in \mathcal{Y} \subset \mathbb{R}^{k \times N}$, and the training data points are drawn independently from an unknown source distribution, i.e., $\boldsymbol{x}_i^0 \sim p_0(\boldsymbol{x})$. Given a deep neural network $f_\theta$ with learnable weights $\theta$, we can solve for an optimal model using the standard empirical risk minimization (ERM): $\hat{\theta}_0 = \arg\min_\theta \sum_i \mathcal{L}(f_\theta(\boldsymbol{x}_i^0), \boldsymbol{y}_i^0)$, where $\mathcal{L}(\cdot)$ is a suitable loss function. If the training dataset is large enough and the network structure is complex enough, the ERM optimal model will generalize well on unseen data points, drawn from $p_0(\boldsymbol{x})$ [39]. The challenge in CL is that the input distribution may be non-stationary. Hence, the testing samples may be drawn from drifted distributions. The resulting distributional gap will lead to poor generalization during the testing time. Our goal is to update the source-trained model $f_{\hat{\theta}_0}$ continually using solely unlabeled data to avoid poor generalization without forgetting past experiences. To model this process,

we consider a set of sequentially arriving target domains $\mathcal{T}^t, t = 1 \dots T$, with unlabeled datasets $\mathcal{D}_\mathcal{T}^t = (\boldsymbol{X}_t)$, where $\boldsymbol{X}_t \in \mathbb{R}^{d \times M_t}$, $\boldsymbol{x}_i^t \sim p_t(\boldsymbol{x})$, and $\forall t_1, t_2 : p_{t_1} \neq p_{t_2}$ (see Figure 1). Since these domains are unlabeled using ERM is implausible. As stated, common UDA methods cannot address CL settings.

To address challenges of "domain shift" and "catastrophic forgetting", we consider that the base network $f_\theta(\cdot)$ can be decomposed into a deep encoder $\phi_{\boldsymbol{v}}(\cdot) : \mathcal{X} \to \mathcal{Z} \subset \mathbb{R}^p$ and a classifier subnetwork $h_{\boldsymbol{w}}(\cdot) : \mathcal{Z} \to \mathcal{Y}$, i.e., $f_\theta = h_{\boldsymbol{w}} \circ \phi_{\boldsymbol{v}}$, where $\theta = (\boldsymbol{w}, \boldsymbol{v})$. Our method is based on consolidating the internal distribution that is formed in the embedding space $\mathcal{Z}$. We assume that as the result of initial training on the source domain, the source classes become separable in $\mathcal{Z}$. If at each time-steps, we update the model such that the internal distribution remains stable, i.e., the distance between the distributions $\phi(p_0(\boldsymbol{x}^0))$ and $\phi(p_t(\boldsymbol{x}^t))$ is minimized, then the model continues to generalize well on the target domains, despite initially being trained with the source domain labeled dataset. This strategy has been used extensively by the existing UDA algorithms but we are constrained with the accessibility of $\mathcal{D}_\mathcal{S}$ in CL, i.e., the term $\phi(p_0(\boldsymbol{x}^0))$ cannot be computed.

## 4   Proposed Solution

Our solution is based on consolidating the intermediate distribution that is learned in the discriminative embedding space to retain model generalizability. Upon learning the initial source domain, the encoder is trained to map the input source distribution into a multi-modal distribution $p_J(\boldsymbol{z})$ in the embedding space. Each mode in this distribution corresponds to one of the input classes and the training data points that belong to a particular class, are mapped to the same cluster (see Figure 1, middle). The internally learned multimodal distribution is represented empirically by the source domain data representations $\{(\phi_{\boldsymbol{v}}(\boldsymbol{x}_i^0), \boldsymbol{y}_i^0)\}_{i=1}^N$. We use a GMM $p_J^0(\boldsymbol{z})$ with $k$ component as the parametric model to estimate the internal distribution:

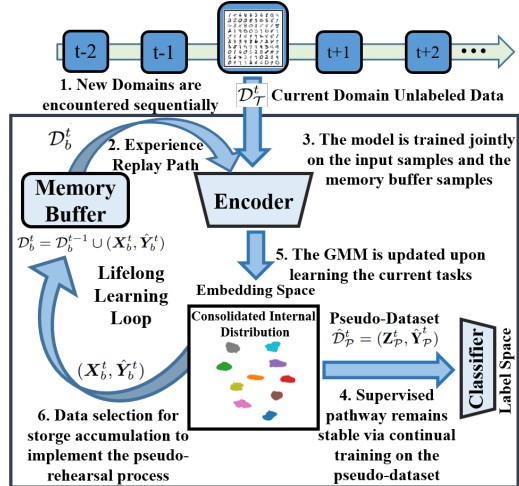

**Lifelong Domain Adaptation Agent**

$$p_J^0(\boldsymbol{z}) = \sum_{j=1}^{k} \alpha_j^0 \mathcal{N}(\boldsymbol{z}|\boldsymbol{\mu}_j^0, \boldsymbol{\Sigma}_j^0), \quad (1)$$

Figure 1: Architecture and learning procedure for the proposed lifelong UDA framework. For an enlarged version, please refer to the Appendix.

where $\alpha_j^0$ denotes the mixture weights, and $\boldsymbol{\mu}_j^0$ and $\boldsymbol{\Sigma}_j^0$ denote the mean and co-variance matrices for the components. Computing these parameters is generally a challenging iterative procedure. But since the labels are accessible, we can compute the parameters for each mode independently. Let $\boldsymbol{S}_j^0$ denote the support set for the mode $j$, i.e., $\boldsymbol{S}_j^0 = \{(\boldsymbol{x}_i^0, \boldsymbol{y}_i^0) \in \mathcal{D}_\mathcal{S}| \arg\max \boldsymbol{y}_i^0 = j\}$. The MAP estimates for the GMM parameters would be:

$$\hat{\alpha}_j^0 = \frac{|\boldsymbol{S}_j^0|}{N}, \ \hat{\boldsymbol{\mu}}_j = \sum_{(\boldsymbol{x}_i^0, \boldsymbol{y}_i^0) \in \boldsymbol{S}_j^0} \frac{1}{|\boldsymbol{S}_j^0|} \phi_v(\boldsymbol{x}_i^0), \ \hat{\boldsymbol{\Sigma}}_j^0 = \sum_{(\boldsymbol{x}_i^0, \boldsymbol{y}_i^0) \in \boldsymbol{S}_j^0} \frac{1}{|\boldsymbol{S}_j^0|} \big(\phi_v(\boldsymbol{x}_i^0) - \hat{\boldsymbol{\mu}}_j^0\big)^\top \big(\phi_v(\boldsymbol{x}_i^0) - \hat{\boldsymbol{\mu}}_j^0\big). \quad (2)$$

We can update the estimates in Eq. 2 for $t > 1$, to be described later. Let $\hat{p}_J^t(\boldsymbol{z})$ denote the estimated GMM at the time-step $t$. Our idea is to consolidate this distribution to retain model generalization. In the absence of source data, we adapt the model such that the encoder always aligns the target distributions with this distribution in the embedding space. We implement this process by drawing random samples from the GMM and build a labeled pseudo-dataset: $\hat{\mathcal{D}}_\mathcal{P}^t = (\boldsymbol{Z}_\mathcal{P}^t, \hat{\boldsymbol{Y}}_\mathcal{P}^t)$, where $\boldsymbol{Z}_\mathcal{P}^t = [\boldsymbol{z}_1^t, \dots, \boldsymbol{z}_{N_p}^t] \in \mathbb{R}^{p \times N_p}$, $\hat{\boldsymbol{Y}}_\mathcal{P} = [\hat{\boldsymbol{y}}_1^{p,t}, \dots, \hat{\boldsymbol{y}}_{N_p}^{p,t}] \in \mathbb{R}^{k \times N_p}$, and $\boldsymbol{z}_i^t \sim \hat{p}_J^t(\boldsymbol{z})$. We determine the labels according to the model predictions. We include only those samples for which the model prediction confidence-level is more than a preset threshold $\tau$ to exclude outlier samples. To retain the model generalization power, we solve the following problem at time $t$:

$$\min_{\boldsymbol{v},\boldsymbol{w}} \sum_{i=1}^N \mathcal{L}\big(h_{\boldsymbol{w}}(\boldsymbol{z}_i^p), \hat{\boldsymbol{y}}_i^{p,t}\big) + \lambda D\big(\phi_{\boldsymbol{v}}(p_t(\boldsymbol{X}_t)), \hat{p}_J^t(\boldsymbol{Z}_\mathcal{P}^t)\big), \quad (3)$$

where $D(\cdot, \cdot)$ denotes a probability discrepancy measure and $\lambda$ is a trade-off parameter. The first term enforces the classifier to preserve its generalizability on the internal distribution. The second term enforces alignment of the target domain distribution with the internally learned distribution in the embedding space. We use the SWD metric [27] for the term $D(\cdot, \cdot)$. Upon updating the model, we can also update the internal distribution $\hat{p}_J^t(\boldsymbol{z})$ using the pseudo-dataset samples.

Solving Eq. (3) would help the model to generalize well on the new domain $\mathcal{T}^t$. However, catastrophic forgetting will remain untackled because the encoder subnetwork is updated unconditionally to minimize the domain discrepancy term in Eq. 3. To tackle catastrophic forgetting, we rely on experience replay. When a task is learned, we select a small subset of the training data points as representative samples, to be stored in a memory buffer. These samples are replayed during model adaptation to mitigate catastrophic forgetting. Possible selection strategies includes mean of features (MoF) [40], ring buffer [41], and reservoir sampling [42]. Existence of the internal distribution makes MoF a natural choice. Upon model update at time-step $t$ and then updating the GMM, we can compute the distance of representations for all the data points of $\mathcal{T}^t$ from their corresponding cluster mean, i.e., $d_{j,l}^t = \|\mu_j^t - \phi(\boldsymbol{x}_l^t)\|_2^2, \forall \boldsymbol{x}_l^t$ s.t. $\hat{\boldsymbol{y}}_l^t = \arg\max f_{\hat{\theta}^t}(\boldsymbol{x}_l^t) = j$. Given a memory budget of $N_b$ samples, we pick the $M_b = N_b/k$ samples per class that have the closest distance to the mean to form the buffer-stored dataset $\mathcal{D}_b^t = \mathcal{D}_b^{t-1} \cup (\boldsymbol{X}_b^t, \hat{\boldsymbol{Y}}_b^t)$. These samples are the most informative samples to estimate the internal distributions. We update Eq. (3) to tackle catastrophic forgetting as follows:

$$\min_{\boldsymbol{v}, \boldsymbol{w}} \sum_{i=1}^{N} \mathcal{L}(h_{\boldsymbol{w}}(\boldsymbol{z}_i^p), \hat{\boldsymbol{y}}_i^{p,t}) + \sum_{i=1}^{N_b} \mathcal{L}(h_{\boldsymbol{w}}(\phi_{\boldsymbol{v}}(\boldsymbol{x}_i^b)), \hat{\boldsymbol{y}}_i^b) + \lambda D(\phi_{\boldsymbol{v}}(p_t(\boldsymbol{X}_t)), \hat{p}_J^t(\boldsymbol{Z}_{\mathcal{P}}^t)) + \lambda D(\phi_{\boldsymbol{v}}(p_t(\boldsymbol{X}_b^t)), \hat{p}_J^t(\boldsymbol{Z}_{\mathcal{P}}^t)).$$

(4)

Adding the second supervised term in Eq. (4) helps to mitigate catastrophic forgetting. The fourth term helps to consolidate the internal distribution across all the previous tasks. Our proposed solution for lifelong domain adaptation, named Lifelong Domain Adaptation Using Consolidated Internal Distribution (LDAuCID), is presented and visualized in Algorithm 1 and Figure 1, respectively.

## 5 Theoretical Analysis

We provide an analysis within a standard PAC-learning framework [39]. Consider the set of classifier sub-networks $\mathcal{H} = \{h_{\boldsymbol{w}}(\cdot) | h_{\boldsymbol{w}}(\cdot) : \mathcal{Z} \rightarrow \mathbb{R}^k, \boldsymbol{v} \in \mathbb{R}^V\}$ as the hypothesis class within PAC-learning. Let $e_0$ denote the expected error on the source domain, $e_t$ denote the expected error on the target domains, and $e_t^J$ denote the expected error on the pseudo-dataset for a given $h \in \mathcal{H}$. Also, let $\hat{p}_0(\boldsymbol{x}) = \frac{1}{N} \sum_{n=1}^{N} \delta(\phi_{\boldsymbol{v}}(\boldsymbol{x}_n^s))$ and $\hat{p}_t(\boldsymbol{x}) = \frac{1}{M_t} \sum_{m=1}^{M_t} \delta(\phi_{\boldsymbol{v}}(\boldsymbol{x}_m^t))$ denote the empirical source and target distributions in the embedding space. We provide the following theorem.

**Theorem 1**: Consider LDAuCID algorithm at learning time-step $t = T$. Then for all the previously learned tasks $t < T$, the following holds:

$$e_t \leq e_{T-1}^J + W(\phi(\hat{p}^t), \hat{p}_J^t) + \sum_{s=t}^{T-2} W(\hat{p}_J^s, \hat{p}_J^{s+1})$$

$$+ e(\boldsymbol{w}^*) + \sqrt{(2\log(\frac{1}{\xi})/\zeta)}\left(\sqrt{\frac{1}{M_t}} + \sqrt{\frac{1}{N_p}}\right.$$

$$+ \sqrt{\frac{1}{N_b}}\Big),$$

(5)

where $e(\boldsymbol{w}^*)$ denotes expected error for the best joint-trained optimal model in the hypothesis

---

**Algorithm 1** LDAuCID $(\lambda, \tau, N_b)$

1: **Source Training:**
2:   **Input:** source labeled dataset $\mathcal{D}_{\mathcal{S}} = (\boldsymbol{X}_0, \boldsymbol{Y}_0)$
3:     $\hat{\theta}_0 = (\hat{\boldsymbol{w}}_0, \hat{\boldsymbol{v}}_0) = \arg\min_\theta \sum_i \mathcal{L}(f_\theta(\boldsymbol{x}_i^0), \boldsymbol{y}_i^0)$

4:   **Internal Distribution Estimation:**
5:     Use Eq. (2) and estimate $\alpha_j^0, \boldsymbol{\mu}_j^0,$ and $\Sigma_j^0$
6:   **Memory Buffer Initialization**
7:     $\mathcal{D}_b^0 = (\boldsymbol{X}_b^0, \hat{\boldsymbol{Y}}_b^0)$
       Pick the $N_b/k$ samples with the least
         $d_{j,l}^t = \|\mu_j^t - \phi(\boldsymbol{x}_l^t)\|_2^2,\ \hat{\boldsymbol{y}}_b^{0,i} =$
    $\arg\max f_{\hat{\theta}^t}(\boldsymbol{x}_b^{0,N_b})$
8: **Continual Unsupervised Domain Adaptation:**
9: **for** $t = 1, \ldots, T$ **do**
10:     **Input:** target unlabeled dataset $\mathcal{D}_{\mathcal{T}}^t = (\boldsymbol{X}_t)$
11:     **Pseudo-Dataset Generation:**
12:       $\hat{\mathcal{D}}_{\mathcal{P}}^t = (\boldsymbol{Z}_{\mathcal{P}}^t, \hat{\boldsymbol{Y}}_{\mathcal{P}}^t) =$
13:       $([\boldsymbol{z}_1^{p,t}, \ldots, \boldsymbol{z}_N^{p,t}], [\hat{\boldsymbol{y}}_1^{p,t}, \ldots, \hat{\boldsymbol{y}}_N^{p,t}])$, where:
          $\boldsymbol{z}_i^{p,t} \sim \hat{p}_J^{t-1}(\boldsymbol{z}), 1 \leq i \leq N_p$ and
          $\hat{\boldsymbol{y}}_i^{p,t} = \arg\max_j\{h_{\hat{\boldsymbol{w}}_t}(\boldsymbol{z}_i^{p,t})\}$ if with
          confidence $\tau$: $\max_j\{h_{\hat{\boldsymbol{w}}_t}(\boldsymbol{z}_i^{p,t})\} > \tau$
14:     **for** $itr = 1, \ldots, ITR$ **do**
15:       draw data batches from $\mathcal{D}_{\mathcal{T}}^t$ and $\hat{\mathcal{D}}_{\mathcal{P}}$
16:       Update the model by solving Eq. (4)
17:     **end for**
18:     **Internal Distribution Estimate Update:**
19:       Use Eq. (2) similar to step 5 above.
20:     **Memory Buffer Update**
21:       $\mathcal{D}_b^t = \mathcal{D}_b^{t-1} \cup (\boldsymbol{X}_b^t, \hat{\boldsymbol{Y}}_b^t)$, where $(\boldsymbol{X}_b^t, \hat{\boldsymbol{Y}}_b^t)$
         is computed similar to step 7 above.
22: **end for**

space, i.e., the model trained as $\boldsymbol{w}^* = \arg\min_{\boldsymbol{w}} e_c(\boldsymbol{w}) = \arg\min_{\boldsymbol{w}}\{e_t(h) + e_J^t(h)\}$, $W(\cdot,\cdot)$ denotes the WD distance, and $\zeta$ and $\xi$ are two constants.

**Proof: Included in the Appendix.**

Theorem 1 explains why LDAuCID algorithm is effective. Major terms in the right-hand side of Eq. 5, as an upperbound for the expected error for each task (domain), are continually minimized by LDAuCID. The first term is minimized because the internal distribution random samples are used to minimize the empirical error term as the first term of Eq. (4). The second term is minimized as the third terms of Eq. (4) when the task distribution is aligned with the empirical internal distribution in the embedding space at time $t$. The third term which is a summation which models the effect of continual learning. The $s^{th}$ term in this sum is minimized when the task $\mathcal{T}^s$ is learned and we deliberately update the internal distribution. This additive term grows as more tasks are learned after learning a particular task which can potentially make the upperbound looser, leading to forgetting effects. The fourth term is a constant small term if the tasks are related, i.e., share the same classes. LDAuCID does not minimize this terms. It just suggests that the model should work well for a joint-training scenario in order to perform well in a sequential learning setting. The last term is a constant term which is negligible if we have sufficient numbers of source and target training data points. It also states that storing more samples in the memory buffer leads to better performance, as expected. We conclude if the upperbound in Eq. (5) is sufficiently tight, the domains are relevant, and the hypothesis space is a suitable space to learn the tasks, then adapting the model using LDAuCID can both tackle catastrophic forgetting and also improve model generalization on the target domains.

# 6 Empirical Validation

Our implemented code is accessible as an Appendix.

## 6.1 Data Sets and Tasks

We are not aware of any prior work that addresses UDA in a lifelong learning setting that we can directly compare against. For this reason, we use four existing UDA benchmark datasets, adopt them to build sequential UDA tasks, and validate our method on these classic datasets. For comparison purpose, we run our algorithm in the learning setting of the existing UDA methods with one source and one target domains. For fair comparison against these works, we have followed the evaluation protocols that are used by most of the recent classic UDA papers.

**Digit recognition tasks:** the common MNIST ($\mathcal{M}$), the USPS ($\mathcal{U}$), and the SVHN ($\mathcal{S}$) datasets are used as three domains. Majority of the existing UDA methods report their results on three tasks defined between these domains: $\mathcal{M} \to \mathcal{U}, \mathcal{U} \to \mathcal{M}$, and $\mathcal{S} \to \mathcal{M}$ tasks. We perform experiments on two $\mathcal{S} \to \mathcal{M} \to \mathcal{U}$ and $\mathcal{S} \to \mathcal{U} \to \mathcal{M}$ sequential UDA tasks to cover the three classic UDA tasks.

**ImageCLEF-DA Dataset:** this dataset consists of the 12 shared image classes between the Caltech-256 ($\mathcal{C}$), the ILSVRC 2012 ($\mathcal{I}$), and the Pascal VOC 2012 ($\mathcal{P}$) visual recognition datasets. The dataset is fully balanced with 50 images per class, i.e., 600 images for each domain. There are six possible binary UDA tasks. We perform experiments on $\mathcal{C} \to \mathcal{I} \to \mathcal{P}$ and $\mathcal{C} \to \mathcal{I} \to \mathcal{P}$ ternary domain tasks.

**Office-Home Dataset**: this a more challenging object recognition dataset which consists of $15,500 images$ in office and home settings. The images are grouped into 65 classes. There are 4 domains with relatively large gaps: Artistic images ($A$), Clip Art ($C$), Product images ($P$), and Real-World images ($R$) which leads to possibility of defining 12 pair-wise binary UDA tasks. We preform experiments on the sequential UDA tasks $A \to C \to P \to R$ and $R \to P \to C \to A$ tasks.

**Office-Caltech Dataset**: this object recognition dataset is built using the 10 shared classes between the Office-31 and Caltech-256 datasets. There are four visual domains: Amazon ($A$), Caltech ($C$), DSLR ($D$), and Webcam ($W$) with 2533 images in total. There are 12 definable binary UDA tasks. We perform experiments on the sequential UDA tasks $A \to C \to D \to W$ and $W \to D \to C \to A$.

## 6.2 Network Structure and Evaluation Protocol:

We use the VGG16 network as the base model for the digit recognition tasks. Following the literature, we use the Decaf6 features for the Office-Caltech tasks. For the other two datasets, we use the

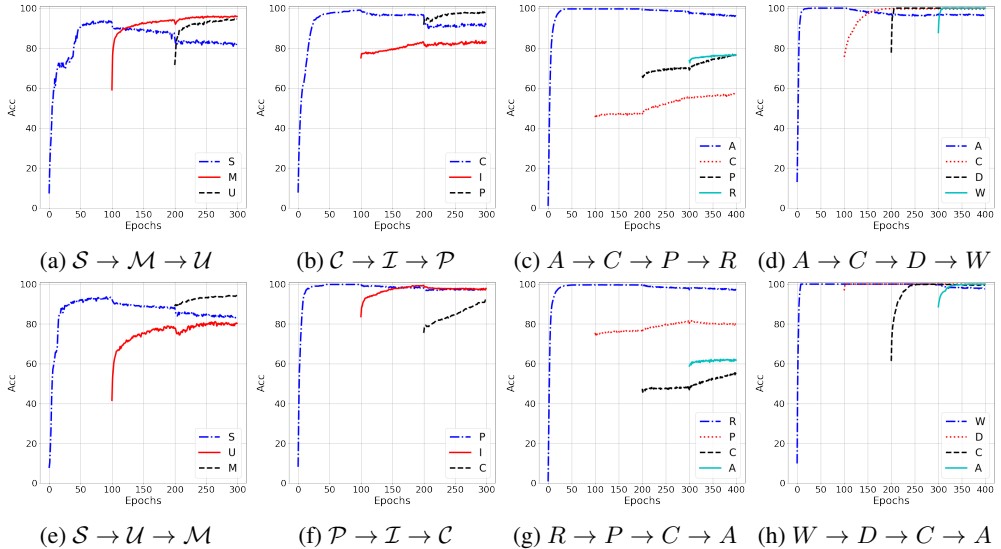

Figure 2: Learning curves for sequential UDA tasks on (a,e) digit, (b,f) ImageClef, (c,g) Office-Home, and (d,h) Office-Caltech datasets. (Best viewed in color).

ResNet-50 network which is pre-trained on the ImageNet dataset as the backbone of the network. To study the model learning dynamics over time, we generate learning curves by reporting the model performance on the testing split of the learned tasks versus the training epochs. Our purpose is simulate continual training during execution time. Providing the learning curves allows to study dynamics of learning. For comparison purpose and after learning a target domain task, we report the average classification rate and the standard deviation on the target domain for five runs for each task. We train the base model using the source labeled data. We report the performance of the model before adaptation as a simple ablation study to demonstrate the effect of domain shift. Then we adapt the model using the target unlabeled data using LDAuCID algorithm and report the performance of LDAuCID on the target domain. For more details on implementation, please refer to the Appendix.

## 6.3 Results

Learning curves for the above eight sequential UDA tasks are visualized in Figure 2. In these learning curves, the model has been trained for 100 epochs, i.e., a notion for time, for each task and then we have moved forward to learn the subsequent task. We have stored 10 samples per class per domain in the memory buffer for experience replay. We first observe that domain shift leads to an initial performance drop on the target domain in all the eight tasks, however there is a jumpstart in performance. That is, the learning curves for the subsequent target tasks initialize significantly higher than random label assignment. For example, we observe in Figure 2a the initial testing accuracies for the domain $\mathcal{M}$ and $\mathcal{U}$ are in the $+60\%$ range. This initial jumpstart occurs due to similarities between the domains, leading to knowledge transfer from past experiences. The rising behavior of the learning curves after this initial performance, when the model is being trained on $\mathcal{T}^t$, demonstrates that LDAuCID effectively improves performance of the source-trained model. This improvement occurs for all the target domain tasks. We note that this the final performance improvements on the Office-Home dataset is less because domain gap between the domains are larger for this dataset.

The second important observation is that catastrophic forgetting has been mitigated quite well. We observe that after moving forward to learn the subsequent domains for all the eight tasks, the model performance on the previously learned tasks is relatively stable and forgetting effect is not dramatic. We note that the forgetting effects for the SVHN dataset in $\mathcal{S} \to \mathcal{U} \to \mathcal{M}$ and $\mathcal{S} \to \mathcal{M} \to \mathcal{U}$ tasks are more severe. This is likely because this dataset is more challenging and hence a larger number of samples in the memory buffer may be required. We conclude that LDAuCID algorithm has successfully improved the model generalization while mitigating catastrophic forgetting.

To measure ability of our algorithm, we also compare LDAuCID against existing UDA. We are not aware of any prior work that addresses UDA in a lifelong learning setting. For this reason, we

| Method | $\mathcal{M} \to \mathcal{U}$ | $\mathcal{U} \to \mathcal{M}$ | $\mathcal{S} \to \mathcal{M}$ | Method | $\mathcal{M} \to \mathcal{U}$ | $\mathcal{U} \to \mathcal{M}$ | $\mathcal{S} \to \mathcal{M}$ |
|---|---|---|---|---|---|---|---|
| GtA [10] | $92.8 \pm 0.9$ | $90.8 \pm 1.3$ | $92.4 \pm 0.9$ | CDAN [13] | 93.9 | 96.9 | 88.5 |
| CoGAN [46] | $91.2 \pm 0.8$ | $89.1 \pm 0.8$ | - | SHOT [47] | $89.6 \pm 5.0$ | $96.8 \pm 0.4$ | $91.9 \pm 0.4$ |
| ADDA [3] | $89.4 \pm 0.2$ | $90.1 \pm 0.8$ | $76.0 \pm 1.8$ | CyCADA [48] | $95.6 \pm 0.2$ | $96.5 \pm 0.1$ | $90.4 \pm 0.4$ |
| RevGrad [15] | $77.1 \pm 1.8$ | $73.0 \pm 2.0$ | 73.9 | JDDA [44] | - | $97.0 \pm 0.2$ | $93.1 \pm 0.2$ |
| DRCN [21] | $91.8 \pm 0.1$ | $73.7 \pm 0.4$ | $82.0 \pm 0.2$ | OPDA [27] | 70.0 | 60.2 | - |
| ETD [49] | $96.4 \pm 0.3$ | $96.3 \pm 0.1$ | $\mathbf{97.9} \pm 0.4$ | MML [50] | 77.9 | 60.5 | 62.9 |
| Source Only | $90.1 \pm 2.6$ | $80.2 \pm 5.7$ | $67.3 \pm 2.6$ | LDAuCID | $\mathbf{96.8} \pm 0.2$ | $\mathbf{98.4} \pm 0.1$ | $91.4 \pm 2.2$ |

Table 1: Classification accuracy for UDA tasks between MNIST, USPS, and SVHN datasets.

employ our algorithm in the special case of having only two domains and compare its performance with the classic UDA algorithms. We consider that only one target domain exists and report our performance on this domain. We still address UDA in a sequential learning setting, but is the best we can do for comparison. For comparison, we included the classic UDA methods: GtA [10], DANN [43], ADDA [3], MADA [11], DAN [14], DRCN [21], RevGrad [15], JAN [16], JDDA [44], and UDAwSD [45]. We include results of these works in case the original paper have reported performance on the corresponding dataset. In the Tables, bold font denotes the best performance.

Comparative results for six bi-domain digit recognition tasks are summarized in Table 1. In each table, we have included the Source Only performance as a baseline. It reports the performance of the source-trained model on the target domain without adapting the model. Despite the fact that the classic UDA methods use the full source dataset for joint-training, we observe that LDAuCID outperforms the UDA methods in two of the tasks and its performance on the remaining task is competitive. We observe that ETD is outperforming LDAuCID and on average possesses the highest performance. This is not unexpected because this method is quite close to our method. ETD relies on a version of WD distance for improvement for classic UDA joint-training scheme.

Table 2 presents the results for the ImageCLEF-DA dataset tasks. We note LDAuCID has lead to a significant improvement on this dataset. As mentioned, this dataset is fully balanced both in terms of number of data points per domain and also per class. Note that in our method we rely on the empirical versions of the distributions for domain alignment (check the Appendix for details). A balanced dataset makes the empirical distribution a less biased approximation of the real distribution. We conclude that balanced datasets across the domains can boost performance of our method.

Table 3 summarizes the comparative results for the Office-Home dataset. We observe that CDAN has the best average performance but LDAuCID is still competitive and on four of the tasks outperforms CDAN. As we discussed, domains of this dataset has larger gap compared to the rest of the datasets we used. This could be verified in Figure 2 from the lower jumpstart performance value. As a result, minimizing the second term of the upperbound in Eq. 5 is more challenging for the Office-Home dataset. Performance of CDAN is higher likely because CDAN aligns the two distributions class-conditionally. We conclude that a possible direction to improve our method further in the future is to benefit from class-conditional alignment techniques, e.g, pseudo-labeling the target domain data.

From inspecting Tables 1–4, we conclude that LDAuCID performs quite competitively on all the UDA tasks, despite addressing lifelong UDA constrains in which most of the source domain data points are not accessible. Although LDAuCID likely needs be upperbounded by classic UDA algorithms in terms of performance, but we observe it outperforms many of these UDA methods on some of the standard UDA tasks. Although our motivation was to address UDA in a CL setting, our observations conclude LDAuCID can be used for addressing classic UDA. We hope that development of subsequent methods for lifelong UDA learning regime would make a more thorough comparison possible.

## 6.4 Analytic and Ablative Studies

To study the effect of the algorithm empirically, we have analyzed the geometry of data points in the embedding space. Data geometry approximates the distributions that are learned in the output of the encoder, i.e., empirical version of the internally learned distribution. For this purpose, we take advantage of the UMAP [54] visualization tool to reduce the dimension of the data representations in the embedding space for 2D visualization purpose. In Figure 3, we have visualized the testing splits of the source domain and the two target domains along with a number of randomly drawn samples of the internally learned GMM distribution for the $\mathcal{S} \to \mathcal{M} \to \mathcal{U}$ digit recognition task. In this figure,

| Method | $\mathcal{I} \rightarrow \mathcal{P}$ | $\mathcal{P} \rightarrow \mathcal{I}$ | $\mathcal{I} \rightarrow \mathcal{C}$ | $\mathcal{C} \rightarrow \mathcal{I}$ | $\mathcal{C} \rightarrow \mathcal{P}$ | $\mathcal{P} \rightarrow \mathcal{C}$ | Average |
|---|---|---|---|---|---|---|---|
| Source Only [9] | $74.8 \pm 0.3$ | $83.9 \pm 0.1$ | $91.5 \pm 0.3$ | $78.0 \pm 0.2$ | $65.5 \pm 0.3$ | $91.2 \pm 0.3$ | 80.8 |
| DANN [43] | $82.0 \pm 0.4$ | $96.9 \pm 0.2$ | $99.1 \pm 0.1$ | $79.7 \pm 0.4$ | $68.2 \pm 0.4$ | $67.4 \pm 0.5$ | 82.2 |
| MADA [11] | $75.0 \pm 0.3$ | $87.9 \pm 0.2$ | $96.0 \pm 0.3$ | $88.8 \pm 0.3$ | $75.2 \pm 0.2$ | $92.2 \pm 0.3$ | 85.9 |
| CDAN [13] | $76.7 \pm 0.3$ | $90.6 \pm 0.3$ | $97.0 \pm 0.4$ | $90.5 \pm 0.4$ | $74.5 \pm 0.3$ | $93.5 \pm 0.4$ | 87.1 |
| DAN [14] | $74.5 \pm 0.4$ | $82.2 \pm 0.2$ | $92.8 \pm 0.2$ | $86.3 \pm 0.4$ | $69.2 \pm 0.4$ | $89.8 \pm 0.4$ | 82.4 |
| RevGrad [15] | $75.0 \pm 0.6$ | $86.0 \pm 0.3$ | $96.2 \pm 0.4$ | $87.0 \pm 0.5$ | $74.3 \pm 0.5$ | $91.5 \pm 0.6$ | 85.0 |
| JAN [16] | $76.8 \pm 0.4$ | $88.0 \pm 0.2$ | $94.7 \pm 0.2$ | $89.5 \pm 0.3$ | $74.2 \pm 0.3$ | $91.7 \pm 0.3$ | 85.7 |
| ETD [49] | 81.0 | 91.7 | 97.9 | 93.3 | 79.5 | 95.0 | 89.7 |
| LDAuCID | $\mathbf{87.8} \pm 1.4$ | $\mathbf{99.1} \pm 0.2$ | $\mathbf{100} \pm 0.0$ | $\mathbf{99.8} \pm 0.0$ | $\mathbf{88.8} \pm 1.0$ | $\mathbf{99.5} \pm 0.3$ | $\mathbf{95.8}$ |

Table 2: Classification accuracy for UDA tasks for ImageCLEF-DA dataset.

| Method | A→C | A→P | A→R | C→A | C→P | C→R | P→A | P→C | P→R | R→A | R→C | R→P | Average |
|---|---|---|---|---|---|---|---|---|---|---|---|---|---|
| Source Only [9] | 34.9 | 50.0 | 58.0 | 37.4 | 41.9 | 46.2 | 38.5 | 31.2 | 60.4 | 53.9 | 41.2 | 59.9 | 46.1 |
| DANN [43] | 45.6 | 59.3 | 70.1 | 47.0 | 58.5 | 60.9 | 46.1 | 43.7 | 68.5 | 63.2 | 51.8 | 76.8 | 57.6 |
| CDAN [13] | 49.0 | 69.3 | 74.5 | **55.4** | 66.0 | **68.4** | **55.6** | **48.3** | **75.9** | **68.4** | **55.4** | **80.5** | **63.9** |
| DAN [14] | 43.6 | 57.0 | 67.9 | 45.8 | 56.5 | 60.4 | 44.0 | 43.6 | 67.7 | 63.1 | 51.5 | 74.3 | 56.3 |
| JAN [16] | 45.9 | 61.2 | 68.9 | 50.4 | 59.7 | 61.0 | 45.8 | 43.4 | 70.3 | 63.9 | 52.4 | 76.8 | 58.3 |
| DJT [23] | 39.7 | 50.4 | 62.4 | 39.5 | 54.3 | 53.1 | 36.7 | 39.2 | 63.5 | 52.2 | 45.4 | 70.4 | 50.6 |
| LDAuCID | **48.3** | **67.4** | **74.1** | 48.7 | **61.9** | 63.8 | 49.6 | 42.1 | 71.3 | 60.3 | 47.6 | 76.6 | 59.4 |

Table 3: Classification accuracy for UDA tasks of Office-Home dataset.

| Method | A→C | A→D | A→W | W→A | W→D | W→C | D→A | D→W | D→C | C→A | C→W | C→D | Average |
|---|---|---|---|---|---|---|---|---|---|---|---|---|---|
| Source Only | 84.6 | 81.1 | 75.6 | 79.8 | 98.3 | 79.6 | 84.6 | 96.8 | 80.5 | 92.4 | 84.2 | 87.7 | 85.4 |
| DANN [43] | 87.8 | 82.5 | 77.8 | 83.0 | **100** | 81.3 | 84.7 | 99.0 | 82.1 | 93.3 | 89.5 | 91.2 | 87.7 |
| MMAN [51] | 88.7 | 97.5 | **96.6** | 94.2 | **100** | 89.4 | **94.3** | 99.3 | 87.9 | 93.7 | **98.3** | 98.1 | 94.6 |
| RevGrad [15] | 85.7 | 89.2 | 90.8 | 93.8 | 98.7 | 86.9 | 90.6 | 98.3 | 83.7 | 92.8 | 88.1 | 87.9 | 88.9 |
| DAN [14] | 84.1 | 91.7 | 91.8 | 92.1 | **100** | 81.2 | 90.0 | 98.5 | 80.3 | 92.0 | 90.6 | 89.3 | 90.1 |
| CORAL [52] | 86.2 | 91.2 | 90.5 | 88.4 | **100** | 88.6 | 85.8 | 97.9 | 85.4 | 93.0 | 92.6 | 89.5 | 90.8 |
| WDGRL [53] | 87.0 | 93.7 | 89.5 | 93.7 | **100** | 89.4 | 91.7 | 97.9 | 90.2 | 93.5 | 91.6 | 94.7 | 92.7 |
| LDAuCID | **99.6** | **100.0** | 86.5 | **96.1** | **100** | **99.8** | 88.5 | **100.0** | 95.7 | **99.3** | 96.4 | **99.8** | **96.8** |

Table 4: Performance comparison for UDA tasks of Office-Caltech dataset.

each point represents one data point and each color represents one of the ten digit classes. Each row of Figure 3 represents the data geometry at the embedding space at the end of time-step $t$. For example, the second row shows data geometry after learning the SVHN and MNIST datasets. This means that if we check the sub-figures vertically for each of the domains, i.e., a column of the figure, we can inspect the effect of learning more tasks versus time. Checking all the columns, we see that upon learning a task, the learned knowledge is retained when the model is adapted in the future, i.e., the next bottom rows, because classes stay relatively separable. This stability suggests that catastrophic forgetting has been continually mitigated when the model is updated. We can also see at the last row, all the domains share the same distribution similar to the internally learned GMM distribution. This observation suggests that our method successfully aligns the distributions to share the same internal distribution. Finally and as an example, if we compare the distribution of the MNIST dataset in the first row versus the second row, we see that as the result of domain adaptation, its distribution in the embedding space becomes more similar to the SVHN dataset (source domain). We conclude that Figure 3 empirically confirms the result that we could deduce analytically from Theorem 1.

Similar to most learning algorithms, setting up the proper values for the hyperparamters is important for optimal performance. We have studied the effect of the values for the hyperparameters $\lambda$ and $\tau$ on the model performance for the binary UDA task $\mathcal{S} \rightarrow \mathcal{M}$ in Figure 3e and Figure 3f to suggest how to tune these parameters for practical usage. We have visualized the average performance with a dark curve. The lighter shaded region around the curve denotes the standard deviation. We observe in Figure 3e that the value for the parameter $\lambda$ does not have a significant effect on performance. This observation is according to expectation because the ERM loss term $\mathcal{L}(\cdot)$ in Eq. (3) is relatively small prior to beginning of the alignment process as a result of pre-training on the source domain. In other words, since the dominant term for optimization is the domain-alignment term in Eq. (3), careful finetuning of the trade-off parameter $\lambda$ is not necessary. We also observe in Figure 3f that when the confidence parameter is $\tau \approx 1$, the model performance on the target domain improves. This accords with our intuition because potential outlier GMM distribution samples cause label pollution which is a challenge for domain alignment in UDA. From the results presented in this section, we can deduce that our method is effective and the empirical observations support our theoretical analysis.

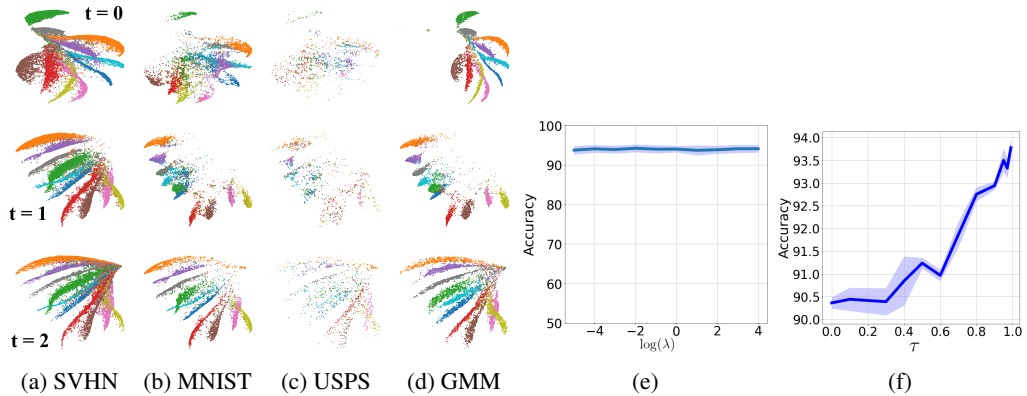

| | t = 0 | t = 1 | t = 2 |
|---|---|---|---|
| (a) SVHN | (b) MNIST | (c) USPS | (d) GMM |

Figure 3: (a-d) UMAP visualization for the testing split of the domains in the UDA task $\mathcal{S} \rightarrow \mathcal{M} \rightarrow \mathcal{U}$ and the fitted GMM at time-steps $t = 0, 1, 2$. Visualizations at each of the rows are computed after learning the $t^{th}$ task prior to the time-step $t + 1$. Enlarged version is included in the Appendix. (e-f) Effect of the values for the hyperparameters $\lambda$ and $\tau$ on model generalization in the target domain for the binary UDA task $\mathcal{S} \rightarrow \mathcal{M}$. (Best viewed in color and on screen.)

| Imbalanced Dataset | | | | $N_b = 50$ | | | | $N_b = 100$ | | | |
|---|---|---|---|---|---|---|---|---|---|---|---|
| Time-step | $t=0$ | $t=1$ | $t=2$ | Time-step | $t=0$ | $t=1$ | $t=2$ | Time-step | $t=0$ | $t=1$ | $t=2$ |
| $\mathcal{S}$ | 89.8 | 84.8 | 80.2 | $\mathcal{S}$ | 92.3 | 83.6 | 82.0 | $\mathcal{S}$ | 93.3 | 83.6 | 83.1 |
| $\mathcal{M}$ | - | 93.2 | 93.0 | $\mathcal{M}$ | - | 94.2 | 95.6 | $\mathcal{M}$ | - | 95.6 | 97.1 |
| $\mathcal{U}$ | - | - | 93.1 | $\mathcal{U}$ | - | - | 91.9 | $\mathcal{U}$ | - | - | 94.6 |

Table 5: Analytic experiments using the $\mathcal{S} \rightarrow \mathcal{M} \rightarrow \mathcal{U}$ digit recognition task.

We have provided additional controlled experiments on the $\mathcal{S} \rightarrow \mathcal{M} \rightarrow \mathcal{U}$ task in Table 5 to draw better intuitions about our proposed algorithm. Since we observed that our algorithm is more effective for balanced datasets, we first study the robustness of our algorithm with respect to data imbalance. To this end, we introduced data imbalance for all domains in the $\mathcal{S} \rightarrow \mathcal{M} \rightarrow \mathcal{U}$ task by considering that for digit $i$, only $\frac{i+1}{10}$ portion of the data points are available. We have reported the observed performance in Table 5, where each row shows performance for each task at the end of learning period for task $t$. We observe that despite reduced performance, our method still addresses catastrophic forgetting and leads to improved performance in the target domains for this imbalanced dataset.

Finally, we note another unexplored aspect in our algorithm is the effect of $N_b$ on performance as we set its value arbitrarily. We set $N_b = 10$ in previous experiments because compared to the sizes of the datasets, $N_b$ would be very small. We studied effect of increasing $N_b$ in Table 5 for $N_b = 50$ and $N_b = 100$. As expected, we observe that using a larger buffer size leads to improvements. In practice, the buffer size should be selected to be as large as possible given the hardware storage limitations.

# 7 Conclusions

We develop an algorithm for domain adaptation in a continual learning setting. The core assumption is that when we train a neural classifier, the input distribution is mapped to an internal distribution, in an embedding space modeled by a neural network hidden layer. Our method is based on consolidating this internally learned distributional such that all learned tasks share a similar distribution in the embedding space. As a result, the model generalizability is retained when new tasks are learned. Additionally, catastrophic forgetting is mitigated using experience replay by storing and then replaying the input samples that are more informative for estimating the internally learned distribution. We used a simple approach to estimate the internal distribution but we foresee that better estimations of the internal distribution can boost the continual learning performance. Future research includes studying the effect the tasks' order on continual learning performance and extensions to incremental learning setting, where new classes can be learned beyond the initial training phase.

## Acknowledgement

We thank the anonymous reviewers whose constructive and continual feedback helped to improve the presentation, clarity, and analysis of the proposed work.

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
