# Appendix: Lifelong Domain Adaptation via Consolidated Internal Distribution

**Mohammad Rostami**
USC Information Sciences Institute
Los Angeles, CA 90007
rostamim@usc.edu

## Abstract

This appendix includes additional information, including a high-level explanations about the proposed learning setting, background on the sliced Wasserstein distance, proof of Theorem 1, and details of experimental implementation.

## 1  High-Level Description of the Proposed Method

In Figure 1, we see the high-level description of the lifelong UDA approach. Lifelong learning is an iterative process in which the model is updated persistently. Upon training on a source domain with labeled data, the input data is transformed into a multi-modal distribution in the embedding space. The classification performance will be decent only if the input distribution is matched with this internal distribution. We use a GMM to approximate this internal distribution. Since the source domain data is labeled, computing the GMM parameters is easy and straightforward. The system is then fielded and then over time, it encounters sequential tasks with drifting distributions which would lead to performance degradation. Our core idea is to update the model such that it generalizes well on these new domains while retaining what has been learned about the past tasks. Hence we need to address both "catastrophic forgetting" and "source-free domain adaptation" in a unified framework.

To tackle the above challenges, we update the model such that the input distribution is always matched to the GMM distribution. Since the internal distribution remains stable, model generalization on the new domains will become possible. We also store a subset of input samples in a memory buffer and replay the back for pseudo-rehearsal to mitigate catastrophic forgetting.

## 2  Proof of Theorem 1

Our analysis is based on the following theorem [1] which is derived for the classic binary UDA:

**Theorem 2:** Consider two classification tasks in two domains with the distributions $p^t$ and $p^{t'}$, and we have $n_t$ and $n_{t'}$ training data points. Let $h_{w^{t'}}$ be a model trained for the source domain, then for $d' > d$ and $\zeta < \sqrt{2}$, there is a constant $N_0$, which depends on $d'$, such that for any $\xi > 0$ and $\min(n_t, n_{t'}) \geq N_0 \max(\xi^{-(d'+2)}, 1)$ with probability at least $1 - \xi$ for all $h$, the following holds:

$$
e_t(h) \leq e_{t'}(h) + W(\hat{p}^t, \hat{p}^{t'}) + e(\boldsymbol{w}^*) +
$$
$$
\sqrt{\left(2 \log(\frac{1}{\xi})/\zeta\right)} \left( \sqrt{\frac{1}{n_t}} + \sqrt{\frac{1}{n_{t'}}} \right),
\tag{1}
$$

where $e(\boldsymbol{w}^*)$ denotes the performance of the optimal joint-trained model.

Theorem 2 provides an upper-bound on performance of a given source-trained model, when used in a given target domain. Note that the theorem is symmetric and the two domains can be shuffled.

35th Conference on Neural Information Processing Systems (NeurIPS 2021)

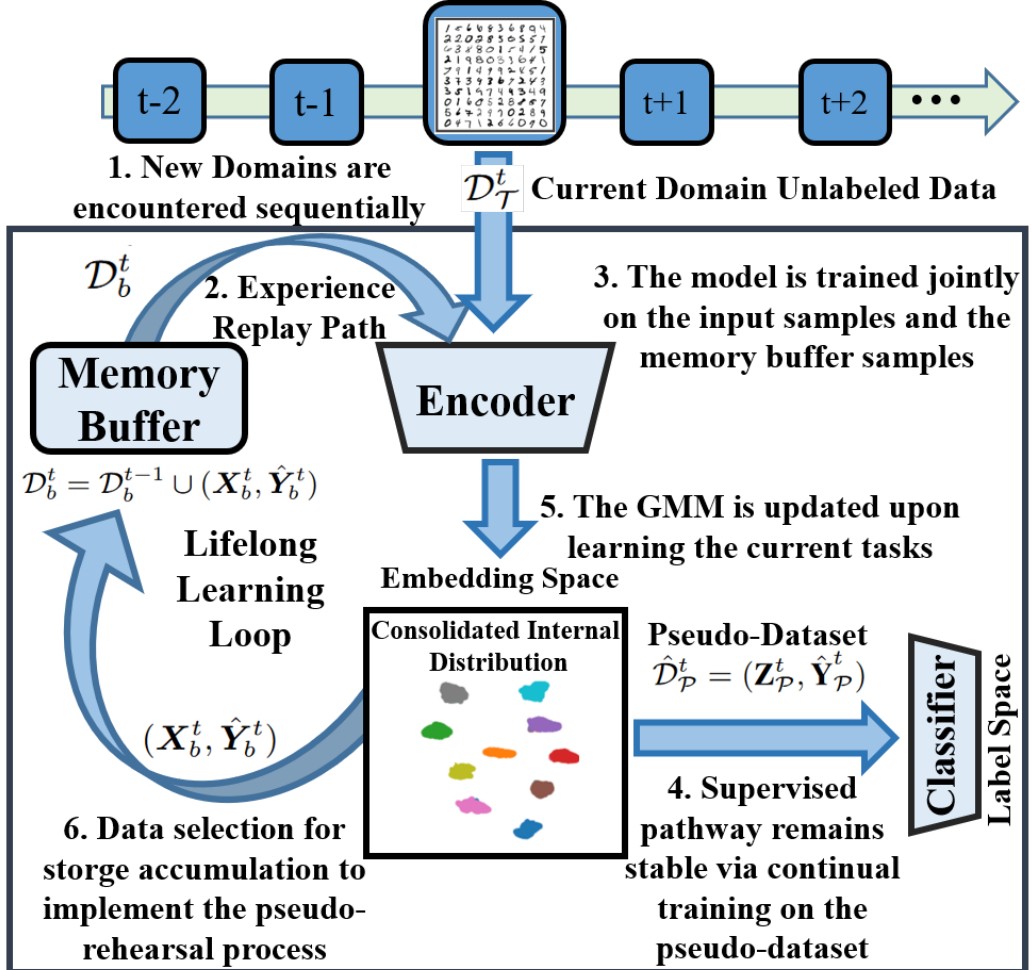

Figure 1: Architecture of the proposed lifelong UDA framework.

It suggests that if the distance between the distributions, when measured in terms of Wasserstein distance, is small and the joint-trained optimal model performance is also has small expected error, then the model performance in the target domain will be similar to the source domain. The third term suggests that the base model should be able to learn both tasks jointly for domain adaptation to be possible. For example, in the case of "XOR classification problem" for binary classification, the tasks cannot be learned by a single model [2]. In other words, the two domains should be relevant. We use this theorem to prove our theoretical result.

**Theorem 1**: Consider LDAuCID algorithm at learning time-step $t = T$. Then fr all tasks $t < T$, we can conclude:

$$
\begin{aligned}
e_t \leq & e_{T-1}^J + W(\phi(\hat{p}^t), \hat{p}_J^t) + \sum_{s=t}^{T-2} W(\hat{p}_J^s, \hat{p}_J^{s+1}) + e(\boldsymbol{w}^*) \\
& + \sqrt{\left(2 \log(\frac{1}{\xi})/\zeta\right)} \left(\sqrt{\frac{1}{M_t}} + \sqrt{\frac{1}{N_p}}\right),
\end{aligned}
\tag{2}
$$

where $e(\boldsymbol{w}^*)$ denotes expected error for the best joint-trained optimal model in the hypothesis space, i.e., i.e., $\boldsymbol{w}^* = \arg\min_{\boldsymbol{w}} e_c(\boldsymbol{w}) = \arg\min_{\boldsymbol{w}}\{e_t(h) + e_J^t(h)\}$. Also, $W(\cdot, \cdot)$ denotes the WD distance, and $\zeta$ and $\xi$ are constants.

**Proof:** Consider the task with the distribution $\phi(p^t)$ as the target domain and also the pseudo-task with the distribution $p_{T-1}^J$ as the source domain in the embedding space in Theorem 2. In Eq. (2), we employ the triangular inequality recursively on the term $W(\phi(\hat{p}^t), \hat{p}_{T-1}^J)$ in Eq. (1), i.e.:

$$W(\phi(\hat{p}^t), \hat{p}_s^J) \leq W(\phi(\hat{p}^t), \hat{p}_{s-1}^J) + W(\hat{p}_s^J, \hat{p}_{s-1}^J), \tag{3}$$

and continue recursion for all time steps $t \leq s < T$. Adding up all the resulting terms, concludes Eq. (2), as desired.

## 3  Details of Experimental Implementation

In the digit recognition experiments, we resized the images of SVHN dataset to $28 \times 28$ images to have the same size of the MNIST and the USPS datasets. This is necessary because we use the same encoder across all domains. Hence, the input data need to share the same size.

In our experiments, we used cross entropy loss as the discrimination loss. At each training epoch, we computed the combined loss function on the training split of data and stopped training when the loss function became constant. We used Keras for implementation and ADAM optimizer. We tune the learning rate such that the training loss function reduces smoothly. We have run our code on a cluster node equipped with 2 Nvidia Tesla P100-SXM2 GPU's. The implemented code is provided as a supplement.

All the datasets have their own standard training/testing splits in all domains. For each experiment, we used these testing splits to measure performance of the methods that we report in terms of classification accuracy. We used the classification rate on the testing set to measure performance of the algorithms. We performed five training trials and reported the average performance and the standard deviation on the testing sets for these trials.

## 4  Enlarged Version of Figure 3

For possibility of better visual inspection, enlarged version of Figure 3 is included.

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

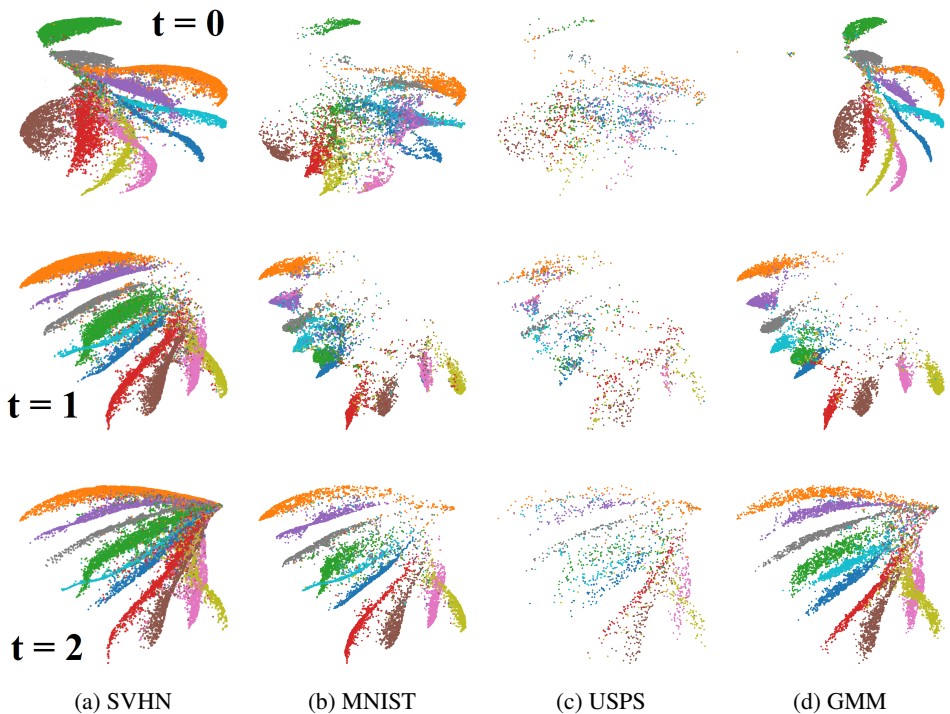

(a) SVHN    (b) MNIST    (c) USPS    (d) GMM

Figure 2: UMAP visualization for the testing split of the domains in the task $\mathcal{S} \to \mathcal{M} \to \mathcal{U}$ and the fitted GMM at time steps $t = 0, 1, 2$: visualizations of each row are computed after learning the $t^{th}$ task prior to the time step $t + 1$. (Best viewed in color and on screen.).