# OpenReview forum: "Lifelong Domain Adaptation via Consolidated Internal Distribution"
_NeurIPS.cc/2021/Conference — NeurIPS 2021 Poster_

### Official Review · Reviewer_RVm3 · 2021-07-15

**Rating:** 7
**Confidence:** 5

**Summary:**

This paper proposes a problem formulation at the intersection between continual learning (CL) and unsupervised domain adaptation (UDA). It introduces a method to perform UDA throughout the lifespan of a model, mitigating catastrophic forgetting of previous information. The proposed method is validated on benchmarks proposed ad-hoc for this problem, as well as on standard UDA benchmarks - showing that is can compete with state-of-the-art approaches even in standard settings.

**Limitations And Societal Impact:**

The theoretical analysis provided hints on the conditions in which the proposed methods will work. Societal impact is not discussed but I agree with the Authors that there is not much to discuss in that regard, for what concerns the submitted manuscript.

**Main Review:**

This is a well written paper on an important topic, that has not received the proper amount of attention yet (CL+UDA). The proposed solution builds on standard techniques typically used in UDA (minimizing distance between domains in the space of learned representations) and in CL (pseudo-rehearsal): they are merged in a very reasonable way, into a method that is empirically shown to perform well on sequences of tasks - where by "task" we intend "UDA problem".

I liked the idea of using GMM for pseudo-rehearsals, with respect to related work on CL that use more complex strategies based on VAEs/GANs: it is cleaner, simpler to implement, and it fits well with the Mean-of-Features memory approach that is used.

My main criticism is that other works have studied CL+UDA problems before, and this work claims to be the first of its kind (I'm reporting the main references below). In particular, [A] proposes a solution that is very similar to the one proposed in this paper. It is originally devised for domains that shift gradually, but it could be easily applied to the benchmarks studied in this paper. Can the Authors comment on this? While [B] is presented as a method for semantic segmentation, the proposed pipeline seems to be fully generalizable to classification problems.

Minor: in the Digits experiment, SVHN is never used as the final domain. MNIST $\rightarrow$ SVHN adaptation is very challenging in UDA, hence it would be interesting to see what happens when SVHN is the final domain in a learning sequence.

I enjoyed reading this paper, and I believe it will help bringing attention to a research direction that is currently a narrow niche, CL+UDA. I did not give a higher score because the related work section - and, in turn, comparisons and discussions - does not include the few works that have addressed the problem tackled in this paper. I look forward reading the Author response on this point, and updating my score accordingly.

References for related work section ([A] and [B] seem the most relevant to me, but the others can further enrich the section)

[A] Bobu et al., Adapting to Continuously Shifting Domains, ICLR Workshops 2018

[B] Wu et al., ACE: Adapting to Changing Environments for Semantic Segmentation, ICCV 2019

[C] Wulfmeier et al., Incremental Adversarial Domain Adaptation for Continually Changing Environments, ICRA 2018

[D] Porav et al., Don't Worry About the Weather: Unsupervised Condition-Dependent Domain Adaptation, ITSC 2019

**Time Spent Reviewing:**

/

---

> ### Author Response · Authors · 2021-08-06
> **Response to the Reviewer Reviewer RVm3**
>
> Thank you for your feedback. We are glad that you have found our work to be novel and on an important unexplored learning scenario.
>
> Thank you for pointing us to the papers we have missed. However, we think our work is different from these works. Please note:
>
> [A] This work is trying to address a similar learning setting to ours but it is not addressing “continual learning” the way that it is defined in the literature. Please note most works for continual learning assume that data for the past learned tasks is not accessible during model execution. However, we see in Eq (4) of [A] that the authors are suggesting is to address the problem of UDA with several target domains by breaking Eq (3) into binary UDA problems. For solving each of these binary domains, source domain data is required, as seen in Eq (4). Hence the authors of A neither claim nor address continual learning, the way most works define it. Hence, this work cannot address our learning setting in its current version. Moreover, the experiments are extremely limited to a relatively simple dataset and it is not clear whether the method would work for the types of benchmarks we have explored.
>
> [B] Again if you check line 9 of the Algorithm in this work, the authors use the source domain samples to update the model, and similar to [A], this work is not addressing continual learning the way that we address.
>
> [C] This work is addressing a practically interesting setting but is not addressing our setting. The primary focus is on addressing gradual shifts.
>
> [D] This work is addressing a particular application which despite being practically interesting, it is highly limited in that it is using image translation. Image translation can be helpful if we have multi-views of the same image, e.g., an image of the same location under different weather conditions in this work. Due to this limitation, this work cannot address the tasks that we have explored.
>
> In conclusion, we don’t think any of the above works are indeed addressing the problem that we are addressing. Our method is broader and relaxes some of the constraints the above have assumed, e.g., accessibility of the source domain data. We will cite them as they are clearly related works, however direct comparison is not trivial. We also respectfully ask you once again to compare our work with these works, especially with A, in terms of being more extensive.
>
> 2. You are absolutely right that the $M\rightarrow S$ task is a challenging task within digit recognition. Please note that we did not intentionally ignore these tasks. We did not address these tasks because most prior works do not use this task and hence meaningful comparison will not be possible. Please also check the discussion on this task in “Unsupervised Domain Adaptation by Backpropagation, Ganin et al” on page 7.
>
>
> We respectfully ask the reviewer to give us a second chance and compare our work against the listed work with more consideration given our explanations. If you find our response convincing, we hope you reconsider your score.

---

> > ### Comment · Reviewer_RVm3 · 2021-08-12
> > **Thank you for the response**
> >
> > The Authors are right in claiming that [A] and [B] consider a more strict scenario where one needs access to the source samples. Essentially, using CL terminology, [A] and [B] rely on rehearsal, whereas the method proposed in the submitted manuscript relies on pseudo-rehearsal, which is desirable from a privacy/memory perspective. I still believe that [A-D] should be thoroughly described: even though the specifics of the formulations are different, these are works at the intersection between CL and UDA: the submitted manuscript is not the first in this niche. I believe that the Authors should properly describe these works and the way they differ form the proposed strategy in the related work section.
> >
> > Concerning the claim "Image translation can be helpful if we have multi-views of the same image": this is not correct. There are many works addressing the problem of "unpaired" image-to-image translation - that is, the translator model does not need access to the same image in different styles/from different views, but only to the two datasets from the two different domains (see, e.g., the CycleGAN model by Zhu et al. ICCV 2017). Yet, I agree that relying on image-to-image translation in the specific framework studied in the submitted work would not be easy, and in general the machinery required is rather complex.
> >
> > Regarding the claim "most prior works do not use this task [M -> S]": there is a variety of papers that test this specific task. I believe Saito et al. ICML 2017 was one of the first, but several more tested it afterwards.
> >
> > I am increasing my score to 7, under the hope that the Authors will make an attempt in comparing the proposed method with the most related work, even if the specific formulations are not fully comparable.

---

> > > ### Author Response · Authors · 2021-08-14
> > > **Thank you for the second feedback**
> > >
> > > Dear reviewer,
> > >
> > > We understand your time constraints and other commitments and thank you for devoting your time and reading our response. We are glad that you have found our response relatively convincing. We would like to add:
> > >
> > > - We agree about your assessment of our statement on "image-to-image translation". Multi-views of the same images are not necessary for training, but we meant the primary idea is that we convert an image from one view to another view. For example, we translate an image in sunny weather to rainy weather.
> > >
> > > - We will check the literature and add the [M -> S] case in our tables. This is relatively straightforward to do. We will also explore the literature more to add comparison against works that we may have missed.
> > >
> > > Best regards,
> > > Our team

---

### Official Review · Reviewer_T4HK · 2021-07-16

**Rating:** 7
**Confidence:** 4

**Summary:**

This paper presents a novel method for unsupervised domain adaptation in continual learning setting using only unannotated data. The proposed method models the internally learned distribution of initial annotated source domain as multimodal distribution and updates it using the learned distributions of subsequent unannotated target datasets to couple all the domains together i.e., consolidated internal distribution.

To minimize catastrophic forgetting, the proposed method also performs experience replay during domain adaptation on target dataset using important representative samples of all historical tasks.

**Limitations And Societal Impact:**

Addressed by authors.

**Main Review:**

Originality:

1. This work proposes a novel method which combines Unsupervised Domain Adaptation and Continual Learning methods in a joint learning framework via consolidated internal distribution.

2. In terms of novelty, the proposed method has the following strengths:
> - Creating internally learned distribution using initial annotated source dataset as multimodal distribution and consolidating it over a sequence of subsequent unannotated target datasets via domain alignment in shared embedding space using Sliced Wasserstein Distance (SWD) metric.
> - Identifying important representative data points for each historical dataset and using them during current dataset's training for knowledge retention of past datasets/domains and to minimize catastrophic forgetting.

---

Quality:

1. The motivation is well founded and the claims are sound.

2. Theoretical formulation and analysis are done in detail and look fine except for the comments below.
> - Step-19 in algorithm 1 denotes update of GMM parameters using newly learned unannotated target dataset $t$, but equation 2 only explains the initialization of GMM parameters using initial annotated source dataset. Therefore, there is a need of another equation for updating GMM parameters using target dataset $t$.
> - See comments regarding inconsistent use of notations below.

---

Clarity:

1. As the proposed method deals with unsupervised domain adaptation and continual learning for classification tasks only, it would be appropriate to mention this in the introduction itself before "costly and time-consuming manual data annotation" statement.

2. In Figure 1, if equations are also referenced in the steps (wherever applicable) as done in algorithm 1, then it should enhance the understanding of the proposed method.

3. A table of notations should improve the readability.

4. Inconsistent use of notations:
> - $N$ is used to signify the number of samples in annotated dataset $D_S$, but $M_t$ is used to signify number of samples in unannotated dataset $D_T^t$ (in line 78).
> - Pseudo-dataset samples are denoted with $z_i^t$ in line 116 and with $z_i^p$ in equation 3 and $z_i^{p,t}$ in algorithm 1.

5. Typographical errors:
> - Line 35: forgetting the CL literature -> forgetting in the CL literature
> - Line 41: update the modal -> update the model
> - Line 72: challenge is CL -> challenge in CL
> - Line 73: non-stationery -> non-stationary
> - Line 75: is update -> is to update
> - Line 134 form form -> form
> - Line 195: Damian -> Domain
> - Figure 2 caption: (a,c) -> (a,e)
> - Table 1 caption: MINIST -> MNIST

6. Methodology:
> 1. As selection of important representative samples for experience replay is done based on the distance to cluster mean in input representation space ($x$) and not in embedding space ($z$), using model prediction confidence scores in addition to distance to cluster mean might help in selecting better representative samples for experience replay.
> 2. While updating GMM parameters, updating parameters using all the samples of target dataset in embedding space might be sub-optimal due to low confidence model predictions. So, it would be interesting to see how model performance varies if parameters are updated based on the samples with model prediction confidence score above a certain threshold hyperparameter.

7. Empirical Evaluation:
> 1. In Figure 2, accuracy of base model trained on each dataset of UDA sequential tasks as initial source dataset can be added and compared with the learning curves of sequential UDA tasks to study the tradeoff between knowledge transfer from past domains and knowledge retention of past domains i.e., minimizing catastrophic forgetting.
> 2. As equation 4 does not contain any hyperparameter to control the intensity of minimizing catastrophic forgetting, the only control parameter is the number of samples $N_b$. So why store only 10 samples per class for experience replay? Is it supported by any ablation study?
> 3. There is no clear description of "Source Only" method shown in all the Tables. If "Source Only" method is training of base model using only initial annotated source dataset then, Figures 2(a,e) both suggest model accuracy after training 100 epochs on source dataset $S$ is around 90. So, why "Source Only" performance in Table 1 in $S \rightarrow M$ column is only 67.3 which is far less than 90?
> 4. Similarly, in Table 2, why "Source Only" performance so much different in columns $C \rightarrow I$ and $C \rightarrow P$ with the same source dataset $C$? However, performance of LDAuCID method on target datasets is similar with columns containing same target datasets.
> 5. A clear indication in Figure 3 (left) regarding which row belongs to which timestep would improve understanding better.

8. Evaluation Setup:
> 1. Why chose max epochs as 100 per task as 100 epochs might not be enough for convergence on initial annotated source dataset which builds the internally learned distribution in embedding space?

---

Significance:

1. Evaluation setup is comprehensive and results show significant improvement in 2 tasks out of 4.
2. This is a significant achievement given the fact that this is the first algorithm which implements Unsupervised Domain Adaptation in Continual Learning setting.
3. Evaluation datasets, tasks and base models have been selected based on the related works in UDA literature.
4. To keep the evaluation fair with other UDA algorithms without continual learning, pairwise tasks have been used.
5. Although, flat learning curves in Figures 2 (c,d,f,g,h) of the initial source dataset during adapting model on subsequent target datasets might indicate that the algorithm puts more weight on minimizing catastrophic forgetting. Hence, there might be a need for a hyperparameter apart from number of samples $N_b$ to control catastrophic forgetting during domain adaptation on subsequent target datasets.

--

Updates:

(24-08-2021): increased score from 6 to 7.

**Time Spent Reviewing:**

14

---

> ### Author Response · Authors · 2021-08-06
> **Response to the Reviewer T4HK**
>
> Thank you for your thorough feedback. We highly appreciate the extensive time you spent on reviewing our work and thank you for the helpful comments and high-quality review you have provided. As the authors, we are impressed that you have detected errors that we have missed to detect. We are glad that you have found our work to be novel. We have provided responses according to your numbering:
>
> Quality.2 : Thank you for pointing out this ambiguity. We agree clarification is necessary and regret not including more details on how we update the GMM. While inspecting, we found a small typo in Algorithm 1 at line 13, where $\hat{p}^{(t)}_J$ should become $\hat{p}^{(t-1)}_J$. Hence, please note that the pseudo-dataset is labeled because when we sample the GMM $\hat{p}^{(t-1)}_J$, computed at the previous task, the corresponding labels are estimated using the classifier subnetwork, as shown in line 13 of Algorithm 1. We use the labeled pseudo-dataset to update the GMM. We will clarify this point in the updated version.
>
> Clarity: Thank you for reading the paper very carefully and providing helpful suggestions. These are important points, yet easy fixes. We will update accordingly to fix them all.
>
> Methodology.1: We regret the typo we have had which has led to confusion but are glad that your comment led us to detect our typo. We have indeed used the distance in the embedding space to select the samples that we store. In other words, $\|\mu_j^t- x_l^t \|_2^2$ should be written $\|\mu_j^t- \phi(x_l^t) \|_2^2$, both in the text and Algorithm 1. Hence, we have indeed followed your suggestion but wrong notations have been used. We hope the update can address your concern.
>
> Methodology.2: We used your suggestion by adding those samples to the pseudo-samples to update the GMM but it did not lead to a meaningful performance improvement. This may not be surprising because when we generate the pseudo-dataset, we already pick the confident pseudo-samples.
>
> Empirical Evaluation.1: This is a good suggestion. We will add the suggested baseline in Figure 2.
>
> Empirical Evaluation.2: Your statement about the control parameter for memory replay is completely correct. Storing 10 samples per class per domain has been arbitrary. We picked this number because compared to the sizes of the datasets we used, this number is very small, and hence using this small number justifies that the memory overload is very small compared to storing the full dataset. Following your comment, we performed a new ablative experiment for the digit task $S\rightarrow M\rightarrow U$, where we performed experiments with $N_b=50$ and $N_b=100$ to consider a larger buffer. The result is reported in the tabular format below:
>
>
> $N_b=50$
>
> 	-------------------------------------------
> 	Time-Step          |    0   |    1   |   2   |
> 	-------------------------------------------
> 	       $ S $       | 92.3   | 83.6   | 82.0 |
> 	------------------------------------------
> 	       $ M $       |   -    | 94.2   | 95.6 |
> 	------------------------------------------
> 	       $ U $       |    -   |    -   | 91.9 |
>
>
> $N_b=100$
>
> 	-------------------------------------------
> 	Time-Step          |    0   |    1   |   2   |
> 	-------------------------------------------
> 	       $ S $       | 93.3   | 83.6   | 83.1  |
> 	------------------------------------------
> 	       $ M $       |   -    | 95.6   | 97.1  |
> 	------------------------------------------
> 	       $ U $       |    -   |    -   | 94.6  |
>
>
> As expected, we observe that using a larger buffer leads to improvements in the results. In practice, $N_b$ should be selected to be as large as possible which is determined by the hardware limitations.
>
>
> Empirical Evaluation.3 and 4: Lack of defining the “source-only” performance has led to the inconsistency you raised. We should and will clarify the “source-only performance” in the text. In short, you are referring to two totally different values in your comment. What “source-only” means in the Tables means that we train the model on the SOURCE domain and without any further training, we test it in the TARGET domain, i.e., the performance of the source-trained model on the target domain. So, the number 67.3 for $S\rightarrow M$ in Table 1, means that we train the model on $S$ for 100 epochs and then without any update, test it on $M$. On the other hand, the value of around 90% that you are referring to, is obtained after training the model on $S$ for 100 epochs and then testing it again on $S$. Hence, the values you mentioned are model performance on different domains. The situation is similar for the ImageClef tasks. We hope this clarifies your concern.
>
> Empirical Evaluation.5: We will update the Figure accordingly.
>
> Evaluation Setup: Broadly speaking, your concern is valid. We used 100 epochs because it already gives competitive results. But for other challenging problems, we agree using more training epochs would be necessary.
>
> We hope that our response can address your primary concerns. If it is the case, our hope is that you reconsider your rating. Irrespective, we think your feedback is helpful to improve our manuscript. Once again, we thank you for the extensive time you have spent and have found your review thoughtful and detailed.

---

> > ### Comment · Reviewer_T4HK · 2021-08-17
> > **Thank you for detailed response**
> >
> > Dear Authors,
> >
> > Thank you for providing detailed clarifications for my concerns/queries.
> > Please find my responses below in the same order.
> >
> > Quality.2: I am still not completely clear on this. Please find below my follow up query.
> >
> > - At time step $t$, the algorithm uses "target unlabelled dataset" ($D_T^t$), "labelled pesudo dataset" ($\hat{D}_P^t$) and "memory-buffer dataset" ($D_b^{t-1}$) to train the complete model using equation 4. As you mentioned, "labelled pesudo dataset" ($\hat{D}_P^t$) is generated using GMM and classifier network at previous time step ($t-1$) and again "labelled pesudo dataset" is used to update GMM at time step $t$. So, to update the GMM after model training is finished at time step $t$, are the label predictions ($\hat{Y}_P^t$) performed again using updated model (classifier network)? If not, then how learning from new domain will be consolidated in GMM? Will this consolidation be done in next time step $t+1$?
> >
> > - In Algorithm-1 (line-15): "memory-buffer dataset" ($D_b^{t-1}$) is missing
> >
> > I look forward to read the author response on my above-mentioned query and update my score.
> >
> > Empirical Evaluation.2: Thank you running additional experiments. This was indeed expected. My suggestion would be to also add "run-time" and "memory usage" at each time step for fair comparison. Also, add these tables in supplementary.
> >
> > Empirical Evaluation.3 and 4: Clear.
> >
> > Evaluation Setup: A higher value of max epochs might be required by looking at the learning curves of $M$, $I$, $U$ and $C$ datasets in figures 2(a), 2(b), 2(e) and 2(f) respectively. But explanation provided by authors is fair enough in the context of this problem.

---

> > > ### Author Response · Authors · 2021-08-18
> > > **Updated Response**
> > >
> > > Dear Reviewer,
> > >
> > > Thank you for your continual engagement in post-rebuttal discussions and for paying attention to details.
> > >
> > > - Regarding updating the GMM, we did not update the label predictions using the updated classifier, and you have a good point about the procedure to consolidate the new domain in the GMM. When a new domain is learned, the model is updated according to Eq (4) to match the distribution of the new domain to the distribution $\hat{p}_J^{t-1}$ through the pseudo-dataset. When this process is done, the distribution of the new domain is matched to the distribution of the pseudo-dataset. In other words, the pseudo-dataset will be a good representative of the distribution for the new domain as well. So, if we use the pseudo-dataset to estimate the GMM again, it will be representative of the distribution of the new domain in the embedding space. We have not thought about the possibility of updating the pseudo-dataset labels but please note this should not change the results dramatically. Because the classifier is updated in Eq (4) to continue to perform well on the labeled pseudo-dataset. In other words, the predicted labels for the pseudo-dataset should not be very different from the initial labels because the pseudo-dataset is part of the training data.
> > >
> > >
> > > - Adding the "run-time" and the "memory usage" is a good suggestion. Including the "run-time" is straightforward but including the real memory usage on hardware may be tricky because this will probably depend on details of code implementation. But we can compute an estimate of the memory usage given the number of stored data points for each scenario.
> > >
> > > We hope our response is clarifying but please don't hesitate to point out in case of ambiguity.
> > > Best,
> > > Our team

---

> > > > ### Comment · Reviewer_T4HK · 2021-08-18
> > > > **Further concerns**
> > > >
> > > > Dear Authors,
> > > >
> > > > Thank you for providing further clarification regarding GMM update promptly.
> > > >
> > > > I partially agree with the comment - "When this process is done, the distribution of the new domain is matched to the distribution of the pseudo-dataset. In other words, the pseudo-dataset will be a good representative of the distribution for the new domain as well. So, if we use the pseudo-dataset to estimate the GMM again, it will be representative of the distribution of the new domain in the embedding space. We have not thought about the possibility of updating the pseudo-dataset labels but please note this should not change the results dramatically.". Please find below my concerns.
> > > >
> > > > - Does it not depends on the difference between the domains of $D_T^{t-1}$ and $D_T^t$ datasets?
> > > >
> > > > - Also, the training might not be perfect and some of the "pseudo dataset" samples that were selected based on high confidence scores using classifier network at time step $t-1$ might produce low confidence labels (or different labels) using updated classifier at time step $t$ after training is finished. However, according to algorithm-1, the "pseudo dataset" generated at time step $t+1$ will use classifier network trained at $t$ and at that time updated labels will be used to update the GMM. But, it is a question of should updated labels be considered at time step $t$ or $t+1$?
> > > >
> > > > - Although representative samples from target dataset $D_T^t$ are stored in memory-buffer for experience replay, high confidence (high label confidence via classifier network) samples from target dataset $D_T^t$ are ignored in updating GMM which might play a negative role in consolidating the distribution from all dataset domains.
> > > >
> > > > Looking forward to author response.

---

> > > > > ### Author Response · Authors · 2021-08-20
> > > > > **Response to the concern**
> > > > >
> > > > > Dear reviewer,
> > > > > Thank you for your continual engagement in constructive discussions. Your comment encouraged us that we may improve our results if a better GMM estimation approach can be used. To this end, we performed a new experiment on the $\mathcal{S}\rightarrow\mathcal{U}\rightarrow\mathcal{M}$ task while changing the GMM update approach. To estimate the GMM for $t\ge 1$:
> > > > >
> > > > > - We first predicted the labels for the pseudo-data points using the updated model.
> > > > >
> > > > > - We also predicted the labels for the unlabeled data points for the current task using the updated model.
> > > > >
> > > > > We then combined both data points and then estimated the GMM. We have reported the new results below (similar to the tables in the above responses):
> > > > >
> > > > >
> > > > > ---------------------------------------------------------
> > > > >      Time Step       |     0     |     1     |     2
> > > > > ---------------------------------------------------------
> > > > >      $\mathcal{S}$   |     93.1  |  89.8     | 86.7
> > > > > ---------------------------------------------------------
> > > > >       $\mathcal{U}$  |     -     |  90.2     | 88.33
> > > > > ---------------------------------------------------------
> > > > >        $\mathcal{M}$ |     -     |     -     | 96.2
> > > > > ---------------------------------------------------------
> > > > >
> > > > >
> > > > > Comparing these performances with those in Figure 2 (e), it seems that your comment is correct and it is possible to boost performance by using more information to estimate the GMM. We want to highlight that this observation does not mean our prior method is wrong, but it shows improvement over what we have done is still possible. We can update the GMM estimation step as above and update the performances in all experiments, but we wanted to address your concern soon so we did not rerun all the experiments. We can definitely do it in a longer time period as only 8 lines of our code have been changed.
> > > > > Best,
> > > > > Our team

---

> > > > > > ### Comment · Reviewer_T4HK · 2021-08-23
> > > > > > **Conclusion**
> > > > > >
> > > > > > Dear Authors,
> > > > > >
> > > > > > Thank you for addressing my concerns by re-running a subset of experiments with suggested setup.
> > > > > > I definitely agree that the original algorithm proposed by the authors is novel and impactful. However, the new experimental results suggest that the suggested approach is in the right direction and there might be a scope of further improvement.
> > > > > >
> > > > > > I put my trust in the authors that they will try to update the methodology, experiments and other section of the paper according to the agreed upon suggestions in the discussion.
> > > > > >
> > > > > > As all of my concerns have been answered appropriately and sufficiently, I stand by my claim and increase my score to 7.

---

> > > > > > > ### Author Response · Authors · 2021-08-24
> > > > > > > **Response to Conclusion**
> > > > > > >
> > > > > > > Dear reviewer,
> > > > > > > We appreciate your continual engagement in careful reviewing. Irrespective of the final outcome, your comments helped us to improve our work. We will apply them in the updated version.
> > > > > > >
> > > > > > > Best,
> > > > > > > Our team

---

> ### Author Response · Authors · 2021-08-13
> **Inquiry about our response**
>
> Dear reviewer,
>
> We really appreciate you for spending a considerable amount of time reviewing our work. Your feedback helped us to improve our work, however, we appreciate it if you let us know whether your concerns have been addressed in our response? We think that your concerns can be addressed decently and our hope is that since you have found our work to be "novel", "sound" and "well founded" and also a "significant achievement", we can address your concern through post-rebuttal back-and-forth discussions so you are convinced to change your final rating.
>
> Thank you,
> Our team

---

### Official Review · Reviewer_22Be · 2021-07-16

**Rating:** 5
**Confidence:** 4

**Summary:**

The paper aims to address an unsupervised domain adaptation (UDA) in the continual learning (CL) setting, specifically the sequential unsupervised domain adaptation task. The learned internal distribution estimated as GMM are consolidated using pseudo-rehearsal to mitigate the domain shift and catastrophic forgetting.

**Limitations And Societal Impact:**

The paper mentions some limitations in the results section. 1. Because of the distribution estimation, the algorithm works well on balanced datasets but the performance degrades when the dataset is unbalanced. (shown in Table 1-2); 2. As mentioned in line 266-268, the method has a possible improving direction to benefit from class-conditional alignment techniques.

**Main Review:**

Clarity: The paper writing and formatting is clear and well organized. However, some grammatical errors and typos are found, e.g., line 72, “The challenge is CL is ...”; line 75, “Our goal is update ...”; line 134, duplicated “form”; line 196, the two sequential tasks are exactly the same. Also, the mathematical notations, especially super/subscripts, in the equations and algorithms have typos and sometimes are not consistent, which cause confusion and need to be corrected, e.g. in Eq(3), the superscript for $z_i$ should be $p$ instead of $t$; $e^J_t$ in line 149 and $e^t_J$ in line 16; $t$ instead of 0 is used as superscripts in Algorithm line 7; Algorithm line 13, the pseudo-dataset size “N” should be $N_p$. Please check the grammar through the paper and clean up the typos in the mathematical notations.

Reproducibility: All the experiments are using public datasets, and the code and experiment settings are provided in the paper and supplements, so the proposed method and experiments are expected to be reproducible.

Originality: The task of the paper is a combination of UDA and CL, and is formed as a sequential UDA task, which is a novel task setting in domain adaptation. The novelty of the method is the internal distribution estimation and pseudo-dataset sampling. However, the loss function and pseudo-rehearsal are existing techniques in UDA and CL and have limited novelty.

Significance: Since this task is a novel setting, limited horizontal comparisons are made between the proposed method and other existing methods. From the results in Table 1-4, the proposed method is at least comparable to other classic UDA algorithms. But since there’s no comparisons with other existing continual learning methods, it’s hard to say whether this method has significant advantages in CL. Moreover, the accumulation of memory buffer limits the proposed method being used in CL in practice, since in reality, lifelong learning will take unlimited data, but you cannot accumulate unlimited representative data in the memory buffer.

Quality: The sequential UDA task is novel, and the challenges such as catastrophic forgetting and domain shift in CL are mitigated from the learning curves in Fig.2 as well as the visualization in Fig.3. The proposed algorithm and loss function are based on existing techniques and have both theoretical and experimental support.
However, I think several important questions are not fully explained in this paper:
1. Although the motivation for improving the domain shift in CL with unlabeled data is valuable and significant, the sequential domain adaptation in the problem statement is simply an ideal situation and not a real task in practice. It’s very hard to find such a situation in real life where multiple domains are learned separately and sequentially. In most CL cases, after initial training on labeled source domain, the upcoming continual unlabeled data points are sampled from the source domain and other target domains in random order and are always mixed together, instead of the training setting in the paper, where only one target domain is trained at each time step and no data points from other domains are included.
2. The paper does not explain how the internal GMM distribution is updated each step after initialization from the source domain. What data samples are used for updating the GMM at each time step? Pseudo-datasets or unlabeled data points from the target domain? Also, in the Algorithm 1, for $t=1$, how to get $p^1_J$ for pseudo-dataset sampling in line 13? I assume $p^1_J$ at $t=1$ should be the same distribution as $p^0_J$ for the source domain.
3. Memory buffer are used and accumulated to store the representative data points from each domain. However, since lifelong continual learning takes unlimited data, the memory buffer will also become extremely large after longtime accumulation. Thus, the memory buffer strategy is not preferred in practice. Also, the paper stores 10 samples per class per domain, but how is this number determined? Since this number is quite important for the performance (obviously larger buffer will have better results), I think some explanation is needed.
4. Although horizontal comparisons are made with other UDA methods in two-domain setting, however, since the motivation of the paper is also about improving CL method, I think some comparisons with other classic CL methods are needed in order to demonstrate the advantage of the proposed method for lifelong continual learning and sequential domain adaptation.
5. As Results explained, “A balanced dataset makes the empirical distribution a less biased approximation of the real distribution.” as shown in Table 2. This shows a limitation of GMM internal distribution estimation: this kind of distribution estimation is sensitive to class-wise balance. So I’m a bit concerned about the performance of the method on unbalanced datasets, which is always the case in practice.
6. Two loss terms are added from Eq(3) to Eq(4), however there’s no ablation study for each loss term in the experiments. More ablation study for the loss is definitely needed in order to show the advantage of adding memory buffer terms in the loss function.

Despite the above questions, the paper presents a valuable task combining both UDA and CL, and domain shift is the main challenge for lifelong unsupervised learning which is mitigated in this paper. So I think the paper lies on the borderline and look forward to the authors' response.

**Time Spent Reviewing:**

2 hours

---

> ### Author Response · Authors · 2021-08-06
> **Response to Reviewer 22Be**
>
> Thank you for your feedback. We are glad that you have found our setting and algorithm novel and valuable. We appreciate you being open to discussion and hope through discussions, we can address your concerns and motivate you to reconsider your ratings. Our response numbers are initially according to the chronological appearance of the concerns in your review and then we move to numbers according to questions you have raised.
>
> 1. Thank you for pointing out the typos. We will correct them and make sure a professional scientific editor at our institution does a pass on the manuscript.
>
> 2. As for the comparison with CL methods, please note that almost all existing CL methods assume that ALL the sequential tasks are labeled. In this sense, existing methods cannot be used for comparison in our setting where the subsequent tasks are unlabeled. This is the reason we did not compare against CL methods. In fact, the infeasibility of comparison with CL methods demonstrates the novelty of our work. We might however have missed some references. If you give us pointers to CL methods that we are unaware of that can address UDA in a continual learning setting, we can include them for comparison.
>
> 3. As for limitations of using a memory buffer, we agree using a buffer faces a number of limitations. However, please note using a memory buffer is highly common in the CL literature. There are ways to mitigate the challenges of using a memory buffer. For example, we can pick a buffer with a fixed size. Each time a new task is learned, we can discard some of the samples in the buffer and replace them with samples from the current task. It is true that if we learn too many tasks, eventually the memory buffer cannot save enough samples for all tasks but please note this is an inherent limitation that we cannot overcome. Practical solutions are not perfect, rather sometimes much better than alternatives. In this sense, we hope you consider that an inherent limitation is not a limitation only for our work.
>
> Quality Questions
>
> 1. We can think of a practical scenario. Consider that you have trained a model and then use it for extended time periods. It is natural to face domain-shift in the input distribution, i.e., you are facing the same domain with the same classes but with a drifted distribution. In that setting, if you collect the data you face during execution, say every 6 months, you can define it as a new unannotated task. Then our setting can be used to autonomously update the model without human intervention. In other words, we can address the challenge of domain-shift over extended time periods.
> Also, please note that our training scheme is similar to existing CL methods in the aspects you mentioned. The samples that are stored in the memory buffer are indeed “samples from the source domain and other target domains in random order and are always mixed together”. All these samples are used for updating the model. That is exactly why we can address catastrophic forgetting.
>
> 2. Thank you for pointing out this ambiguity. We agree with your assessment that “The paper does not explain how the internal GMM distribution is updated each step after initialization from the source domain” and regret not including more details on how we update the GMM. While inspecting, we found a small typo in Algorithm 1 at line 13, where $\hat{p}^{(t)}_J$ should become $\hat{p}^{(t-1)}_J$. Hence, please note that the pseudo-dataset is labeled because when we sample the GMM $\hat{p}^{(t-1)}_J$, computed at the previous task, the corresponding labels are estimated using the classifier subnetwork, as shown in line 13 of Algorithm 1. We use the labeled pseudo-dataset to update the GMM. As for computing $p_J^1$, your comment is correct that it is the distribution computed after learning the source domain. We will clarify this point in the updated version.
>
> 3. As we explained above, we agree using a memory buffer has its own limitations. But please note the other primary CL approaches, including using a generative model, weight consolidation, and model expansion also have their own limitations. In other words, none of the solutions for CL is perfect. These limitations are inherent and we can use none of these methods to address all continual learning challenges perfectly but they are much better than not using them. In this sense, we respectfully ask the reviewer to consider that the raised challenge is not specific to our method and all CL methods face the same challenge, including breakthrough works, e.g., “Human-level control through deep reinforcement learning” published in Nature.
> We agree that storing 10 samples per class per domain is arbitrary. We picked this number because compared to the sizes of the datasets we used, this number is small, and hence using this small number justifies that the memory overload is very small compared to storing the full dataset. At the same time, we could get good results with this small number. We agree that selecting this number depends on the memory constraints and also how challenging the problems are. We will add further explanations for more clarification.
>
>
> 4. As we described, we are not aware of a CL work that can address UDA in a continual learning setting. Please note this is not a weakness of our work, rather it shows our work is addressing an unexplored learning scenario.
>
> 5. What we meant was that our method is more effective when the datasets are balanced, rather than saying our method is not applicable for imbalance cases. What we meant was that our method is more effective when the datasets are balanced. Please note the Office-Home and Office-Caltech datasets are not fully balanced and our method still leads to relatively competitive performance. To address your concern further, we performed a new experiment on the $S\rightarrow M \rightarrow$ task, considering an imbalance setting for the datasets. We considered that for digit $i$, only $\frac{(I+1)}{10}$ portion of the data points are retained for the experiment, e.g., for digit 7, we keep $80%$ of the data points randomly. By doing so, we get highly imbalanced datasets. We get the following performances (since we couldn’t use Figures here, we have reported performances in a Tabular form. Each row shows performance for each task at the end of the task, denoted on top of the table. In other words, if you check Figure 2, we are reporting the learning curve values at 100, 200, and 300 epochs):
>
>
>
> 	-------------------------------------------
> 	       Time-step   |    0   |    1     |    2    |
> 	-------------------------------------------
> 	       $ S $       |  89.8  |   84.8   |   80.2   |
> 	------------------------------------------
> 	       $ M $       |   -    |   93.2   |   93.0   |
> 	------------------------------------------
> 	       $ U $       |    -   |    -     |   93.1   |
>
> We can see from the above performances values that even when the datasets are imbalanced, our method still addresses catastrophic forgetting and leads to improved performance in the target domains.
>
>
> 6. Thank you for your thoughtful comment. We have missed this potential to perform an ablative experiment. To address the raised concern, we performed additional experiments. Unfortunately, it is not possible to include a Figure similar to Figure 2 in the paper. So, we report our new ablative study results below in a numeric format similar to the above presentation. We performed three ablative experiments by solving Eq 6 using the digit tasks. We dropped the second ($2^{nd}$) and also the fourth term ($4^{th}$) individually, and then both ($C$). For each case, we solved the simplified version of Eq 6. The results are as follows:
>
>
> 	$2^{nd}$ term is dropped
>
> 	-------------------------------------------
> 	Time-Step          |    0   |    1   |   2    |
> 	-------------------------------------------
> 	       $ S $       | 93.2   | 88.5   | 83.4 |
> 	-------------------------------------------
> 	       $ M $       |   -    | 94.5   | 96.2 |
> 	------------------------------------------
> 	       $ U $       |    -   |    -   | 94.4 |
>
>
> 	$4^{th}$ term is dropped
>
> 	-------------------------------------------
> 	Time-Step          |    0   |    1   |   2    |
> 	-------------------------------------------
> 	       $ S $       |   92.7 |   82.7 |  70.3  |
> 	------------------------------------------
> 	       $ M $       |   -    |  95.7  |   96.1 |
> 	------------------------------------------
> 	       $ U $       |    -   |    -   | 94.3   |
>
>
>
> 	Both terms are dropped
>
> 	-------------------------------------------
> 	Time-Step          |    0   |    1   |   2   |
> 	-------------------------------------------
> 	       $ S $       | 93.1   | 68.1   | 40.0 |
> 	------------------------------------------
> 	       $ M $       |   -    | 95.3   | 95.9 |
> 	------------------------------------------
> 	       $ U $       |    -   |    -   | 94.0 |
>
> We draw informative conclusions from inspecting the first row in the above tables. We can clearly see that when we drop the $4^{th}$ term, catastrophic forgetting is severe for this task in subsequent time-steps, compared to dropping the $2^{nd}$ term. When we drop both terms, we face more severe catastrophic forgetting. This observation accords with our intuition because the $4^{th}$ term is more important for consolidating the internal distribution. We appreciate your comment and will add this experiment in the final draft or Supplementary material.

---

> ### Author Response · Authors · 2021-08-13
> **Inquiry about Our Response**
>
> Dear reviewer,
>
> We understand your time constraints and appreciate you for spending time reviewing our work. Given the borderline nature of your initial rating, we appreciate it if you let us know whether your concerns have been addressed in our response? Our hope is that since you have found our work to be "novel" and "valuable" and also have expressed openness in considering our response, we can address your concern through post-rebuttal back-and-forth discussions so you are convinced to change your final rating.
>
> Thank you,
> Our team

---

> ### Author Response · Authors · 2021-08-23
> **Request for updates**
>
> Dear reviewer,
>
> Thank you for your initial review and the suggestions that helped to improve our manuscript. We are hopeful that since you mentioned you are looking forward to reading our response, you can offer your updated opinion after reading our response. A huge advantage of the current reviewing process is that it allows for continual discussions between the reviewers and the authors to make the process more elaborate and based on more thorough considerations. We hope that you can provide us your take on our response so we can continue our discussions further.
>
> Best,
> Our team

---

> ### Author Response · Authors · 2021-08-26
> **Concluding Period**
>
> Dear reviewer,
>
> The deadline for this phase is concluding. We have tried to address the concerns you raised in our response. If there are still unclear points to you, please kindly let us know. We are very glad to further discuss them during the remaining time.
>
> Best,
> Our team

---

### Official Review · Reviewer_7ykJ · 2021-07-20

**Rating:** 7
**Confidence:** 3

**Summary:**

Unsupervised Domain Adaptation (UDA) is studied in a task sequential continual learning setting. The source domain is assumed to have supervised data and several target domains without labels are being sequentially adapted to. An algorithms is engineered for the specific case of supervised classification source datasets which can be neatly clustered in some high-level representation; then, learning on related (target) datasets which can be classified in the same (source) classes can be done sequentially and without the use of labels. Catastrophic forgetting is mitigated using a standard experience replay approach, but also through the assumptions of the conceptual framework, i.e. strong overlap between source and target classes, which is known to be the easier setting in CL works. Empirical validation is done primarily in the using classic domain adaptation benchmarks, with the difference of using a single source domain for multiple target domains in a sequential fashion. The theoretical analysis and ablation study round up the work, including a discussion of algorithm hyper-parameters.

**Limitations And Societal Impact:**

I do not believe the discussion of limitations is sufficient. I would suggest improvements and clarifications along these lines:
* Consolidating internal distributions from datasets which are very different seems counterintuitive; even if the target dataset is a subset of source dataset, say source=ImageNet, target=Dogs++, there are at least 100+ overlapping classes, but it is at least possible that aligning distributions may lead to problems. This should be discussed in the main text.
* Even if continual UDA is possible, it is not entirely clear whether it is needed or beneficial for performance improvements compared to target task specific  source -> target UAD experiments with tuned hyper-parameters.
* Self-supervised learning on data rich source datasets (e.g. ImageNet++) seems to be a strong alternative, certainly for the datasets where the proposed method was demonstrated; I can agree that there are cases where such transfer is not helpful, e.g. (I suppose) skin cancer classification, but such domains are not explored.

**Main Review:**

The novelty of the approach is reduced in terms of continual learning research, especially due to assumptions about source and target datasets, hence I need to primarily assess the merits of the work based on domain adaptation literature. While labels are not used for the target datasets, the assumption of identical class set is very strong.

The empirical evaluation datasets used here, with domains in the hundreds of examples, make it difficult to imagine what the method would do on more realistic current image data streams. Real datasets are not this neat and definitely not this small, even supervised datasets. It is also hard to separate the improvements due to ImageNet pre-training and those due to specific domain adaptation algorithms. I do not believe it is enough to say that this is what domain adaptation works use, but it may be necessary to educate the readers (and indeed the reviewers) on why these datasets are still challenging and why algorithmic improvement is needed; for example, with large scale source dataset learning (even unsupervised source learning with e.g. SimCLR(v1 or v2)), freezing such models and learning linear output layers is enough for learning many other datasets. Why do we need to keep learning from small source datasets? Why not focus on domains where there is no choice but unsupervised domain adaptation?


I need further clarifications for the following:
* What is the upper bound of performance? Multitask training on both source and target domains using target labels and an empirically weighted joint objective?
* When is "aligning internal distributions" possible or desirable? Many datasets have very different classes, e.g. MNIST and Flowers. Is aligning distributions useful there?
* How robust is the method when there is data imbalance between source classes in target datasets?



Misc
* Could you please explain what is meant in rows 30-31? Why are existing UDA algorithms not suitable for continual learning?
* Row 41: modal or model?
* Row 42: could you please explain what is meant by coupled learned distributions?
* Row 195: Damian or domain?

**Time Spent Reviewing:**

2

---

> ### Author Response · Authors · 2021-08-06
> **Response to Reviewer 7ykJ**
>
> Thank you for your feedback. We think some of your concerns go beyond solely our work and are targeted towards the whole field of domain adaptation. While they may be valid, we respectfully ask the reviewer to check the relevant literature and evaluate our work against the precedent, rather than raising concerns that are standard assumptions in the field. Our responses below are chronologically ordered according to the order you have expressed in your review:
>
> 1. It seems your primary concern is that in the context of “domain adaptation literature”, we have used the strong assumption of “identical class”. While we agree this is a strong assumption, please note that the vast majority of the domain adaptation works (including the ones referenced in the manuscript) use this assumption. We respectfully ask the reviewer to check the recent works on domain adaptation, presented in top-venues like NeurIPS, and compare our work side by side to see that we have used a standard assumption that has been used extensively in the prior work. While this is a strong assumption, please note we have followed the precedent. Moreover, this assumption is realistic in some conditions, despite being strong. For example, also mentioned in the manuscript, when we face distributional drifts in the input classes in a continual learning setting, assuming that the same classes exist in two domains is natural.
>
> 2. When it comes to the experimental setup, e.g., datasets, base networks, features, etc., again please note we have followed the precedent and have used the standard benchmarks. Please note that for a fair comparison against prior works, we have to follow the precedent so our results can be compared against prior work. It is true that the setup is not as challenging as the practical problems, but is the standard setup for research purposes.
>
> 3. Your suggestion, i.e., using a frozen encoder trained using SimCLR and training the output layer which is similar to fine-tuning, is not applicable in domain adaptation settings because to train the very last layer of the network, we still need labeled samples. Please note that in a domain adaptation setting, there are no labeled samples in the target domain to train the weights for the last layer. We agree with you that it is more helpful to use a large-scale source domain, but please note that this will depend on practical logistics that are beyond your control, i.e., what annotated dataset is readily accessible. It is highly possible that you may not have a larger-scale annotated source domain. In those situations, you either need to annotate data and create a large dataset or use what is accessible, e.g., a smaller annotated dataset. However, if you want to annotate data, then it probably makes more sense to directly annotate the target domain data. Hence, fine-tuning is very helpful but not always practical.
>
> 4. Your understanding of the upper bound is correct.
>
> 5. You are right that aligning distributions of two different datasets don’t make sense. That is exactly one of the reasons that in domain adaptation, we assume the classes are shared (see above) to avoid matching unrelated domains. Our theoretical analysis also explains this necessity.
>
> 6. What we meant was that our method is more effective when the datasets are balanced, rather than saying our method is not applicable for imbalance cases. What we meant was that our method is more effective when the datasets are balanced. Please note the Office-Home and Office-Caltech datasets are not fully balanced and our method still leads to relatively competitive performance. To address your concern further, we performed a new experiment on the $S\rightarrow M \rightarrow U$ task, considering an imbalance setting for the datasets. We considered that for digit $i$, only $\frac{(i+1)}{10}$ portion of the data points are retained for the experiment, e.g., for digit 7, we keep $80%$ of the data points randomly. By doing so, we get highly imbalanced datasets. We get the following performances (since we couldn’t use Figures here, we have reported performances in a Tabular form. Each row shows performance for each task at the end of the task, denoted on top of the table. In other words, if you check Figure 2, we are reporting the learning curve values at 100, 200, and 300 epochs):
>
>
> 	-------------------------------------------
> 	       Time-Step   |    0   |   1    |   2   |
> 	-------------------------------------------
> 	       $ S $       | 89.8   |   84.8  | 80.2 |
> 	------------------------------------------
> 	       $ M $       |   -    |  93.2   | 93.0 |
> 	------------------------------------------
> 	       $ U $       |    -   |    -    | 93.1 |
>
> We can see from the above performances values that even when the datasets are imbalanced, our method still addresses catastrophic forgetting and leads to improved performance in the target domains.
>
>
> 7. Existing methods for UDA cannot address continual learning because the data points for all tasks should be accessible simultaneously. This is not the case with continual learning, where only the current task data is accessible. In other words, we learn one task at a time and then move to learn the next task without the possibility of going back to learn that task again. Please also note that these assumptions and constraints are not set up by us and what is considered in the relevant literature.
>
> 8. By coupling distributions, we meant aligning them in the embedding space.
>
> 9. Thank you for pointing out the typos (domain and model). We will correct them.
>
> 10. Regarding whether continual learning is “needed or beneficial”, please note continual learning is a very active research area with extensive interest in the ML community. The reason is that there are practical situations where data points for previously learned domains are not accessible. Please note that you are presuming that you always have access to training datasets after using them, whereas this may not be the case in practice. In the UDA setting, if the source domain data is always accessible and you have labeled data in the target domain, you definitely can train a better model than what we have proposed but the very point of our work is that these are not always the conditions you face in reality. Both continual learning and domain adaptation are considering scenarios that the standard machine learning setting that cannot be used.
>
> 11. Regarding using self-supervised learning, please note SSL can help with generating descriptive features, but we will still need labeled data to train a classifier from the feature space to the label space. Both SSL and UDA are helpful but choosing which one to use depends on the challenges we face in the specific application. If the challenge is that we have a pre-trained model, where source domain data is not accessible, and also we want to adapt that model to generalize in a target domain with NO labeled data, SSL is not applicable. SSL is not a very practical solution for continual learning because.
>
> While we understand time constraints and appreciate your time, we would like to respectfully ask the reviewer to check the relevant papers on UDA and continual learning when it comes to evaluating our assumptions and experimental setup, assess our work against the precedent and reconsider the final rating if our work is found consistent with what is practiced in the literature.

---

> > ### Comment · Reviewer_7ykJ · 2021-08-29
> > **Increasing my score to recommend acceptance (7)**
> >
> > Thank you for thoroughly addressing my concerns, and thank you to the other reviewers for their useful feedback!
> >
> > I have initially evaluated (and scored) this paper from the point of view of its direct contribution to CL research in terms of proposed algorithm, which I still believe is simply making use of basic approaches and isn't too novel or general. But now I view the paper as a good challenge to CL efforts since it takes a novel first step to join the two fields. Could new CL methods do better in the particular case of Continual UDA? This paper asks this question with sound motivation and supports it with the basic work needed to facilitate further progress and followups.
> >
> > Furthermore, the paper does leave some important issues on the table which could stimulate and inspire future CL work, e.g. relaxing some of the stringent assumptions of UDA, such as good class set matches and somewhat balanced data. These restrictions, instead of being a weakness of the setting, show that some continual learning issues remain despite the strong assumptions, such as sequence order dependence, and may be studied here before going to the more general cases, for example S -> M -> U vs. S -> U -> M, it looks like U isn't learned well in the second case. I encourage the authors to make it clear that, while some CL issues are resolved, challenges for CL approaches remain and may be studied in the proposed settings.
> >
> > I thank the authors for their patience and I have increased my score to recommend acceptance (7).

---

> > > ### Author Response · Authors · 2021-08-29
> > > **RE: Increasing my score to recommend acceptance (7)**
> > >
> > > Dear reviewer,
> > > Thank you for putting additional time to read our responses and all discussions with other reviews. We understand you have other commitments and highly appreciate it.
> > > We agree with your comment that our work has not addressed some important challenges and we hope to work on addressing them in our future work. Your comment on studying the order of tasks is right on spot and overall it deserves exploration in the CL literature. We will highlight this fact and add a short discussion.
> > > Best,
> > > Our team

---

> ### Author Response · Authors · 2021-08-26
> **Concluding Period**
>
> Dear reviewer,
>
> The deadline for this phase is concluding. We have tried to address the concerns you raised in our response. If there are still unclear points to you, please kindly let us know. We are very glad to further discuss them during the remaining time.
>
> Best,
> Our team

---

### Author Response · Authors · 2021-08-06
**General Note to the reviewers**

Dear Reviewers,
We appreciate your time and effort to review our work. We are glad to receive a high-quality and extensive set of reviews. We are also glad that the collective sentiment about our work is positive and the majority of the reviewers agree that our work is novel. We think that the majority of the concerns can be addressed straightforwardly. Several of the raised points are quite helpful and indeed helped us to improve the manuscript, e.g., correcting important notational typos, informative ablative studies, and improving the presentation. We have tried to make our individual answers self-contained so small portions of our responses are repetitive because similar concerns have been raised twice. We understand the time constraints but respectfully ask the reviewers to engage in discussion and help us further to improve our work.
Thank you,
Our team

---

### Decision · Program_Chairs · 2021-09-27

**Decision:**

Accept (Poster)

**Comment:**

The reviewers identified a number of strengths and weaknesses in this submission. The authors response led to a detailed discussion, over the course of which some of the reviewers increased their scores. Ultimately, three of the four reviewers recommend acceptance, with the four still being in the borderline range. I see not reason to overrule this vote and therefore recommend acceptance.